# Seasonal advance of intense tropical cyclones in a warming climate

Kaiyue Shan[1], Yanluan Lin[2], Pao-Shin Chu[3], Xiping Yu[4✉] & Fengfei Song[5,6✉]

Intense tropical cyclones (TCs), which often peak in autumn[1,2], have destructive impacts on life and property[3–5], making it crucial to determine whether any changes in intense TCs are likely to occur. Here, we identify a significant seasonal advance of intense TCs since the 1980s in most tropical oceans, with earlier-shifting rates of 3.7 and 3.2 days per decade for the Northern and Southern Hemispheres, respectively. This seasonal advance of intense TCs is closely related to the seasonal advance of rapid intensification events, favoured by the observed earlier onset of favourable oceanic conditions. Using simulations from multiple global climate models, large ensembles and individual forcing experiments, the earlier onset of favourable oceanic conditions is detectable and primarily driven by greenhouse gas forcing. The seasonal advance of intense TCs will increase the likelihood of intersecting with other extreme rainfall events, which usually peak in summer[6,7], thereby leading to disproportionate impacts.

Tropical cyclones (TCs) are one of the most devastating natural disasters in the world. Understanding potential changes in TC activity in response to global warming is important for TC-related disaster prevention[8,9]. Global warming caused by human activities is estimated to be about 1.0 °C above pre-industrial levels, with most of the warming occurring since the mid twentieth century[10]. This warming may have already impacted TC activity on the global scale[11–13]. Although changes in the annual number of global TCs are controversial[11,13,14], an increasing trend in the number of global intense TCs has been noted[12,14], and significant efforts have been devoted to reducing uncertainties in data quality issues[15,16]. Poleward migration of the locations of TC activity was observed in most ocean basins in recent decades and is particularly robust in the western North Pacific basin[17–19], which is related to changes in TC seasonality[20]. A recent study[21] identified an earlier onset of the beginning time of TC season, in terms of the initial formation dates, in the North Atlantic basin associated with a warming ocean in spring, but there is no seasonal change in the date that Atlantic accumulated cyclone energy reaches its 10, 50 or 90% threshold. Overall, detection and attribution of changes in TC activity are among the top priorities of TC research[22].

Intense TCs (TCs with a lifetime maximum wind speed greater than 110 kt) that pose the greatest threat require special attention[23,24]. Most regions lack resilience to intense TCs, which could cause many deaths and damage to property through destructive winds, storm surges, heavy precipitation and inland flooding[25–27]. For these reasons, it is crucial to understand changes in intense TC characteristics[28,29]. Although it is still challenging for high-resolution climate models to capture intense TCs[29–31], statistical analysis suggests that intense TCs are likely to be more susceptible to anthropogenic warming compared with other TC categories[12,28,32]. Changes in many characteristics of intense TCs under a warming climate (for example, the number[12], intensity[28]

and lifespan[32]) are fairly well studied; however, little is known about changes in the seasonal cycle of intense TCs.

In general, intense TCs occur more frequently in autumn than in summer because they require sufficient heat from the ocean to develop[1,2]. The seasonal cycle of intense TCs lags behind that of other high-impact weather events (for example, extreme rainfall events produced by the summer monsoon system), which often peak in summer and are largely determined by the seasonal cycle of atmospheric energy driven by solar radiation[6,7]. Recently, the compound hazards of intense TCs and other high-impact weather events have attracted increasing attention[33,34]. The devastating impact of the compound hazards is well beyond any one of these events individually. They can induce substantial inland rain and flooding associated with multiple weather systems, causing large-scale failures of power and transportation systems, straining emergency responses and depleting disaster preparation resources[3–5]. As the preferred time of occurrence of intense TCs and other high-impact weather events is usually off by one season, the likelihood of their simultaneous occurrence is generally assumed to be small; however, given the seasonal advance of intense TCs, as shown in this study, its potential change should be considered.

## Seasonal advance of intense TCs

We begin with a broad view of the change in the seasonal cycle of intense TC occurrence. The occurrence time of an intense TC is defined by the date on which the TC first achieves its lifetime maximum intensity. TC data during 1981–2017 are taken from the advanced Dvorak Technique–Hurricane Satellite (ADT-HURSAT) dataset ('TC observations'). Figure 1a,b shows the seasonal distribution of intense TC numbers (solid line) and the linear trend (bar) for each month in two hemispheres. There is an obvious seasonal advance of intense TCs in both

[1]State Key Laboratory of Hydroscience and Engineering, Department of Hydraulic Engineering, Tsinghua University, Beijing, China. [2]Department of Earth System Science, Ministry of Education Key Laboratory for Earth System Modeling, Institute for Global Change Studies, Tsinghua University, Beijing, China. [3]Department of Atmospheric Sciences, School of Ocean and Earth Science and Technology, University of Hawai'i at Mānoa, Honolulu, HI, USA. [4]Department of Ocean Science and Engineering, Southern University of Science and Technology, Shenzhen, China. [5]Frontier Science Center for Deep Ocean Multispheres and Earth System and Physical Oceanography Laboratory, Ocean University of China, Qingdao, China. [6]Laoshan Laboratory, Qingdao, China. ✉e-mail: yuxp@sustech.edu.cn; songfengfei@ouc.edu.cn

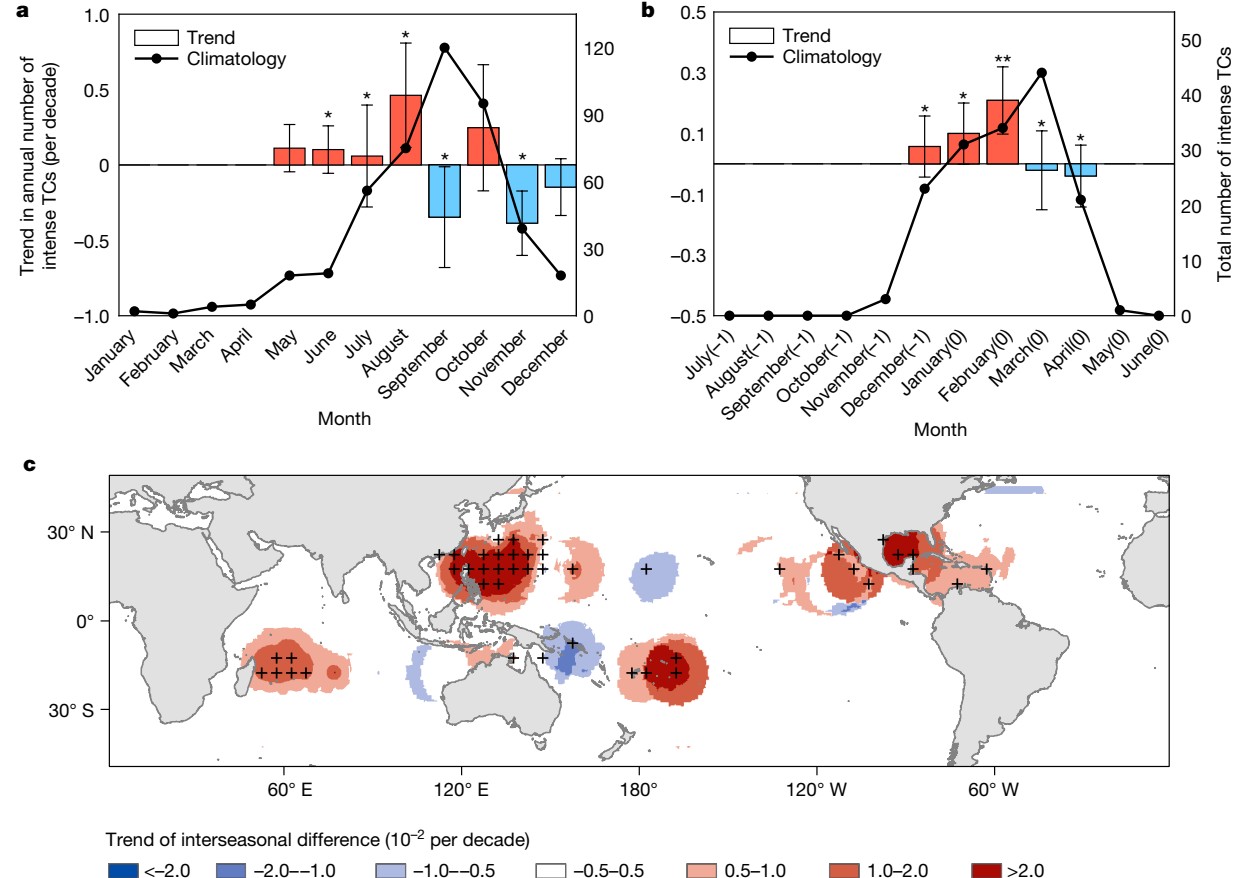

**Fig. 1 | Observed changes in the seasonal cycle of intense TC number.**
**a**,**b**, The seasonal distribution of the total number of intense TCs (solid lines) and the linear trend in the annual number of intense TCs (bars; per decade) obtained from the ADT-HURSAT dataset in each month from 1981 to 2017 in the NH (**a**) and the SH (**b**). On the *x* axis in **b**, −1 indicates the previous year and 0 indicates the current year. *Significance at the 95% confidence level; **significance at the 90% confidence level, based on a non-parametric statistical test ('Statistical significance'). The error bars indicate the 95% confidence intervals in the linear regression analysis. **c**, Linear trends of the interseasonal difference in intense TC number between the early and late seasons obtained from the ADT-HURSAT dataset. The early season is defined as June to August in the NH and December to February in the SH. The late season is defined as September to November in the NH and March to April in the SH. The black crosses indicate significance at the 95% confidence level. The annual number of intense TCs is calculated over 5° × 5° boxes prior to computing trends. The basemap in **c** was plotted using the Matplotlib basemap toolkit with the geographical coordinate system World Geodetic system 1984 generated by the Global Positioning System (maintained by the National Oceanic and Atmospheric Administration).

the Northern Hemisphere (NH) and the Southern Hemisphere (SH) that manifests as increased intense TC occurrence in the early season and decreased intense TC occurrence in the late season ('Occurrence time').

To depict the spatial pattern of the seasonal advance of intense TCs, we estimate the linear trend of the interseasonal difference (early season minus late season) in the number of intense TCs (Fig. 1c). The increasing trends (that is, shifts towards earlier onset) are apparent over the western North Pacific basin, east of 130° W in the eastern North Pacific, over the Gulf of Mexico and the western part of the North Atlantic, west of 80° E of the South Indian, near the northern coast of Australia and east of 180° of the South Pacific basin. Note that the significant earlier-shift trend of intense TC occurrence in the North Atlantic basin features large earlier shifts in the Gulf of Mexico and western part of the North Atlantic basin and nearly no change in the eastern part, which is associated with changes in oceanic conditions[35] and the rarity of intense TC occurrence in the eastern part (Extended Data Fig. 1). The decreasing trends (that is, shifts towards later onset) are observed over the central Pacific and near the northeastern coast of Australia. The consistently earlier-shifting trends of intense TCs over most of the tropical oceans indicate the robustness of this phenomenon.

The time series of the median value of intense TC occurrence time during the active season in two hemispheres based on the ADT-HURSAT

dataset are shown in Fig. 2a,d. In both the NH and the SH, the occurrence time of intense TCs shows a significant trend towards earlier onset at the 95% confidence level based on a non-parametric statistical test ('Statistical significance'), with rate of 3.7 and 3.2 days per decade, respectively (Table 1). The earlier-shifting trend of intense TC occurrence can also be identified in five major ocean basins, and it is significant at the 95% confidence level for each basin except the eastern North Pacific. It is most evident in the western North Pacific basin, with a seasonal advance of 8.1 days per decade, which may have profound impacts on the surrounding regions. This earlier-shifting trend does not depend on how it is estimated or which dataset is used (Table 1).

To confirm whether the seasonal advance only occurs in the intense TCs, we quantify the shifting rates of TCs with different intensities. Using the ADT-HURSAT dataset and best-track dataset ('TC observations'), the results consistently show a significant earlier-shifting trend for intense TCs in two hemispheres based on a non-parametric statistical test but not for less intense TC events (Fig. 2b,e and Extended Data Fig. 2). Note that the earlier-shifting trend in the seasonal occurrence of TCs with intensity larger than 110 kt in the NH obtained from the ADT-HURSAT dataset is statistically significant on the basis of two different statistical tests ('Statistical significance'), whereas the earlier-shifting trend in the SH is statistically significant based on a non-parametric statistical test

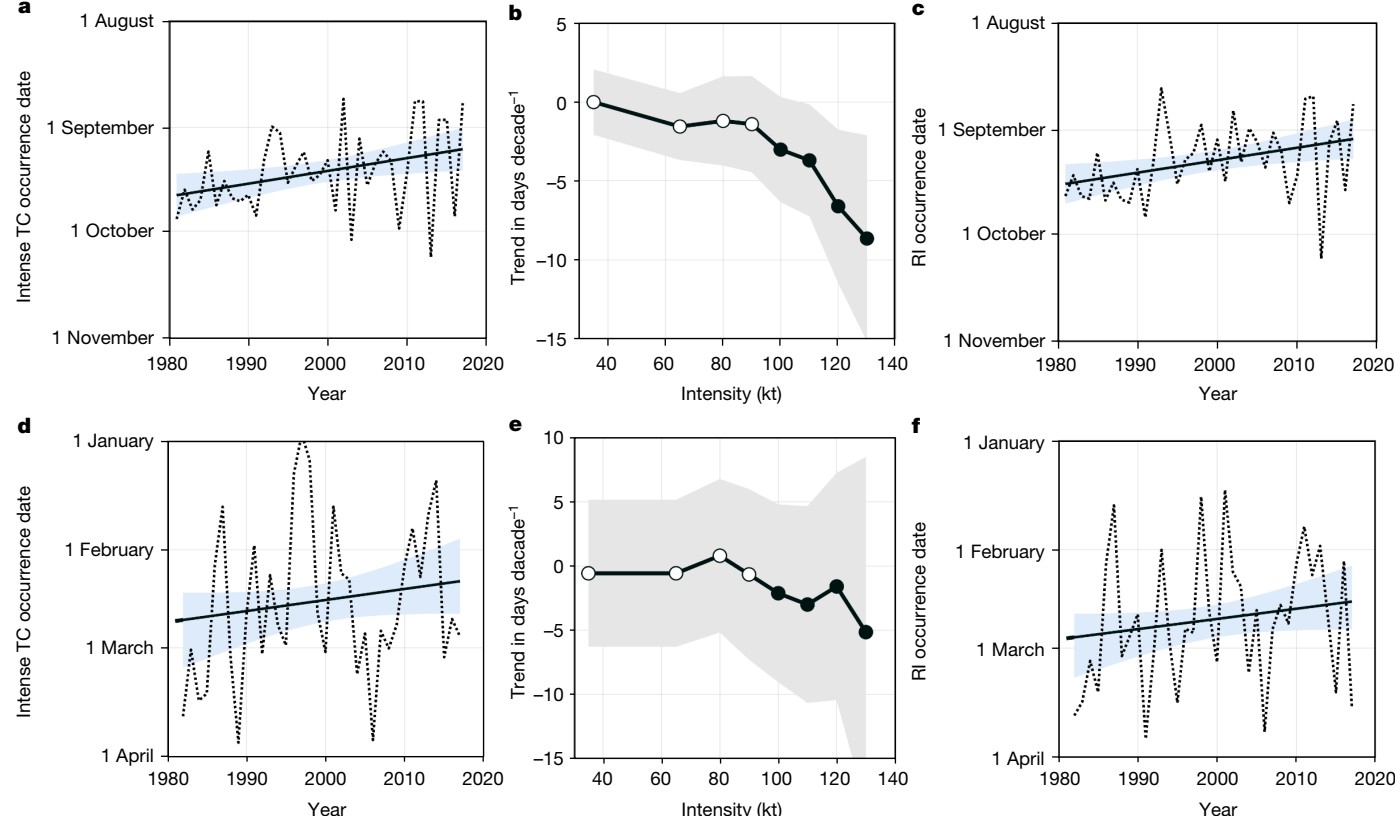

**Fig. 2 | Earlier shifting trends of intense TC occurrence and RI occurrence.** **a**,**d**, Time series of the median value of intense TC occurrence date (dashed lines) and its trend (solid lines) obtained from the ADT-HURSAT dataset in the active season during 1981–2017 in the NH (**a**) and the SH (**d**). The active season is defined as June to November in the NH and December to April in the SH. Linear trends are significant at the 95% confidence level based on a non-parametric statistical test. Shaded areas represent the 95% confidence intervals in the linear regression analysis. **b**,**e**, Linear trends of the median value of the occurrence date (days per decade) of TC events with different intensities obtained from the ADT-HURSAT dataset during the active season in the NH (**b**) and the SH (**e**). The dots denote the significant shifting trends at the 95% confidence level based on a non-parametric statistical test, and the open circles correspond to the insignificant trends. **c**,**f**, Time series of the median value of RI occurrence date (dashed lines) and its trend (solid lines) obtained from the ADT-HURSAT dataset in the active season during 1981–2017 in the NH (**c**) and the SH (**f**).

but with large 95% confidence intervals. The earlier-shifting trends in the NH and the SH obtained from the best-track dataset are statistically significant based on different statistical tests. These results suggest that the seasonal advance is more evident for more intense TCs. In contrast, no significant change is observed in either hemisphere if all TCs are considered (Extended Data Fig. 3a,b). This contrast between intense TCs and less intense ones naturally draws our attention to the essential differences between them.

## Seasonal advance of rapid intensification events

Intense TCs generally experience rapid intensification (RI) in their lifespan[22,36]. In contrast to most other natural phenomena, the probability density function of the lifetime maximum intensity for all TCs is bimodal, with the first maximum around 50 kt and the second one around 120 kt. The existence of the second maximum indicates that there may be a special mechanism that causes a TC to intensify after its intensity exceeds a certain threshold, thus inducing more intense TCs than expected[12,28]. Reference 36 found that the bimodal distribution of TC lifetime maximum intensity could be divided into two unimodal distributions when all TC events were classified into two groups according to whether they experienced an RI process or not. It was also found that the vast majority of intense TCs underwent the RI process at least once during their life cycles and that only a few TCs that never experienced the RI process would develop into intense TCs. It is thus clear that the RI process is the key that distinguishes intense TCs from other TCs.

A natural question, therefore, is whether a similar earlier shift can be found for the seasonal cycle of RI events, which are defined as events with an increase of at least 35 kt over a 24 h period (approximately the 95th percentile of 24 h intensity changes) ('RI events').

As expected, a clear earlier shift in the seasonal cycle of RI events is observed (Extended Data Fig. 3c,d), showing an increase in RI events in the early season and a decrease in RI events in the late season, which is consistent with the earlier shift in the seasonal cycle of intense TC occurrence (Fig. 1a,b). Furthermore, the median value of occurrence dates of RI events shows a significant earlier-shifting trend in both hemispheres (Fig. 2c,f), with rates of 3.6 and 4.1 days per decade in the NH and the SH, respectively. There is a close relationship between the occurrence dates of intense TCs (Fig. 2a,d) and RI events (Fig. 2c,f), with correlation coefficients of 0.88 in the NH and 0.79 in the SH. This highly consistent shifting trend identified between intense TCs and RI events suggests that the earlier shift in the seasonal cycle of intense TCs is related to the earlier shift in RI events.

## Role of seasonally dependent ocean warming

To investigate the reason for the earlier shift in RI occurrence, we examine the seasonal cycles of the oceanic and atmospheric conditions ('Environmental factors') favourable for the RI of TCs[37–40]. For the oceanic conditions, potential intensity (PI)[41] and ocean heat content (OHC) must be very high for RI to occur[8,38]. The former is the theoretical maximum intensity that a TC can achieve and is mainly moderated by

**Table 1 | Linear trends of intense TC occurrence**

| | | Trends (days per decade) | |
|---|---|---|---|
| | | **Median** | **μ** |
| ADT-HURSAT | NH | −3.7*±3.5 | −3.4*±2.9 |
| | SH | −3.0*±7.0 | −3.1*±3.4 |
| Best track | NH | −2.4*±2.2 | −2.7*±2.6 |
| | SH | −3.8*±4.2 | −1.8*±3.5 |

Linear trends of intense TC occurrence dates are calculated based on the ADT-HURSAT dataset during the period 1981–2017 and the best-track dataset during the period 1981–2021. The median value of intense TC occurrence time is calculated over the active season (June to November in the NH and December to April in the SH). The expected value (μ) of the occurrence time is calculated by assuming that the seasonal cycle of intense TCs throughout the TC season approximately follows the beta distribution. *Statistical significance of the linear trend at the 95% confidence level based on the Mann–Kendall test. Negative trends represent earlier shifts. The 95% confidence intervals of the linear trend estimate are shown.

sea surface temperature[41–43]. For the atmospheric part, high atmospheric relative humidity (RH) is beneficial for TC development[44]. The vertical wind shear (VWS) should not be excessively large; otherwise, the TC development process is suppressed[45].

To determine the effect of the oceanic and atmospheric factors on the earlier onset of RI events, the seasonal cycle of each factor and its change are investigated based on the European Centre for Medium-Range Weather Forecasts (ECMWF) and fifth-generation global atmospheric reanalysis (ERA-5) datasets (Fig. 3a–d and Extended Data Fig. 4a–d). It is confirmed that high percentiles of oceanic factors are beneficial for the RI process ('Environmental factors'). An enhanced increase in the fractional area covered by high PI and that covered by high OHC in the early season compared with the decrease in the late season is observed (Fig. 3a–d). This is also consistent among different datasets, increasing confidence in our results (Extended Data Figs. 5 and 6). The earlier onset of favourable oceanic conditions bears a strong resemblance to changes in the seasonal cycle of RI events (Extended Data Fig. 3c,d) as well as intense TCs (Fig. 1a,b). Although atmospheric factors (that is, RH and VWS) were previously documented to also impact RI events and are widely applied as forecasting variables[37,40,45], the seasonal cycle of the atmospheric factors does not show a significant change based on three atmospheric reanalysis datasets (Extended Data Fig. 4). Hence, the observed earlier onset of the favourable oceanic conditions contributes most to the earlier onset of RI events and thus, that of intense TCs, whereas changes in the atmospheric conditions contribute little.

An earlier onset of favourable oceanic conditions is detectable in simulations from global climate models. Figure 3e–h shows the linear trends of the fractional area covered by high PI and that covered by high OHC in two hemispheres over the period 1981–2014. The trends are estimated based on the multimodel mean of the historical simulations from the Coupled Model Intercomparison Project phase 6 (CMIP6) ('Model simulations'). An earlier shift in the seasonal cycle of favourable oceanic conditions for RI occurrence is clear, with an increase in favourable oceanic conditions in the early season and a decrease in the late season. As shown in Extended Data Fig. 7, the earlier shift in the seasonal cycle of favourable oceanic conditions is also detected in the Community Earth System Model v.2 (CESM2) large ensemble, by which internal variability could be largely suppressed ('Model simulations').

Whether the detected earlier-shifting trend in favourable oceanic conditions is driven by anthropogenic or natural forcing remains unknown. To address this question, we compared the contributions of the greenhouse gases (GHGs), natural forcing and anthropogenic aerosol based on the Detection and Attribution Model Intercomparison Project (DAMIP) in CMIP6, which conducted the individual forcing simulations in the historical period ('Model simulations') experiments. Under GHG forcing (Fig. 3i–l), there is an increase in favourable oceanic conditions in the early season and a decrease in the late season.

In contrast, no significant change in the seasonal cycle of favourable oceanic conditions is found under natural forcing (Extended Data Fig. 8a–d). Under anthropogenic aerosol forcing (Extended Data Fig. 8e–h), the fractional area covered by high PI exhibits an increase in March (late season) in the SH, and the fractional area covered by high OHC exhibits an increase in November (late season) in the NH and an increase in March (late season) in the SH, which slightly offset the earlier-shifting trends induced by GHG forcing. These results suggest that the earlier onset of favourable oceanic conditions is primarily driven by GHG forcing, whereas the contributions of the natural forcing and anthropogenic aerosols are negligible or even negative. The earlier shift in the seasonal cycle of the favourable oceanic conditions for RI events is projected to further amplify at the end of this century following a high-emission scenario (Extended Data Fig. 9a–d). This earlier shift trend is consistent with the enhanced sea surface temperature during summertime found in previous studies[44], which is caused by the weakened surface wind in summer[44,46].

## Overlap with extreme rainfall

Our findings have direct implications regarding the risk management of TC-related disasters in a warming climate. The earlier shifting trend of intense TC occurrence is most evident in the western North Pacific basin, which is home to the most TCs on Earth. To investigate its impact on the surrounding regions, South China is taken as an example. In South China, extreme rainfall events exhibit a double-peak annual cycle (Fig. 4a) as revealed by many studies[47,48], with the first peak (in June) influenced by the summer monsoon system and the second peak (in October) induced by TC landfall. An obvious increase in extreme rainfall events is observed during the gap period between the two peaks (July to September). This may be related to the pronounced earlier shift of intense TC occurrence over the western North Pacific basin (Fig. 1). To confirm this, the extreme rainfall events induced by intense TCs ('Extreme rainfall events') were calculated and found to contribute to almost all of the increase in the extreme rainfall in July and half of the increase in August. Consistently, in the Gulf of Mexico, another region heavily impacted by TCs[28,49], an increasing trend in extreme rainfall events during July to September is observed associated with an increasing trend in extreme rainfall produced by intense TCs (Fig. 4b). The seasonal changes in extreme rainfall events in South China and the Gulf of Mexico provide notable examples of increased overlapping periods between the non-TC-related extreme rainfall events and intense TCs. Furthermore, the earlier seasonal occurrence of extreme rainfall produced by intense TCs in the two regions is accompanied by an increasing trend of the annual number of persistent rainfall events ('Extreme rainfall events') as shown in Extended Data Fig. 10, which was reported to bring up disproportionate impacts through inducing a superimposed elevation of water levels and a shortage of disaster preparation resources[5,50]. This motivates us to investigate the simultaneous occurrence of intense TCs and other high-impact weather events quantitatively in the future and account for its risk in planning building infrastructure.

## Discussion

We identified a seasonal advance of observed intense TCs during the recent decades in both hemispheres. The median value of the occurrence time of intense TCs exhibits a statistically significant trend towards earlier onset in the NH and the SH, with rates of 3.7 and 3.2 days per decade, respectively. The results obtained herein are consistent regardless of how the linear trends are estimated and whether data from different sources are used, thus ensuring the robustness of this phenomenon. Interestingly, no significant change is observed for the overall TCs. The discrepancy in the changes between intense TCs and less intense TCs could be largely explained by the earlier shift in the seasonal occurrence of RI events, which is considered the key process

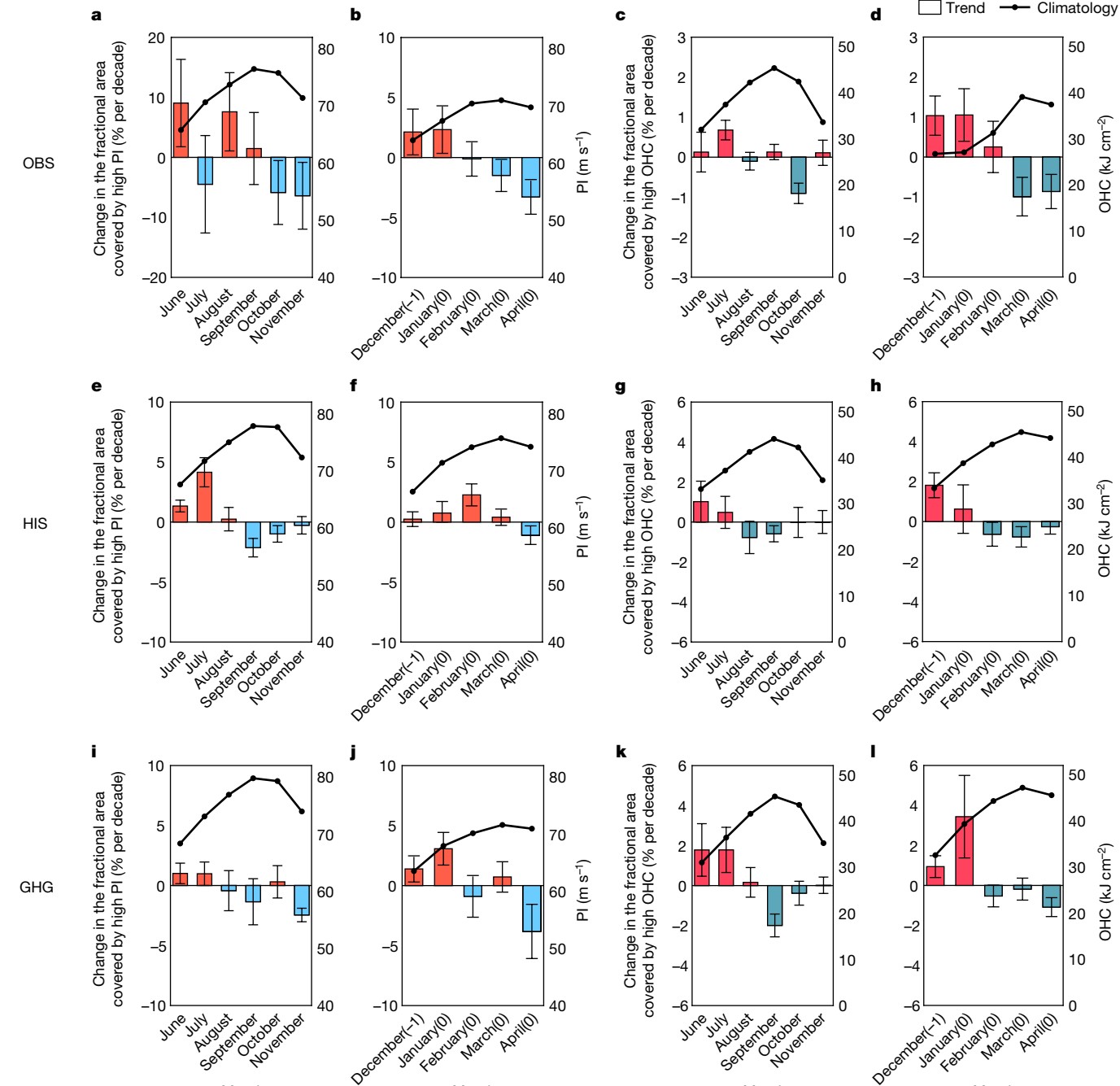

**Fig. 3 | Observed and simulated changes in the seasonal cycle of oceanic conditions.** Observed results (OBS) obtained from the ECMWF and ERA-5 datasets, results from the multimodel mean of CMIP6 historical (HIS) simulations and results from the multimodel mean of simulations forced by GHG only are shown. **a,b**, The climatological mean value of PI (solid lines) and the linear trend of the fractional area covered by high PI (bars) ('Environmental factors') in each month during the active season during 1981–2017 in the NH (**a**) and the SH (**b**). The calculation of PI is based on the oceanic variables from the ECMWF dataset and the atmospheric variables from the ERA-5 dataset. The error bars represent the 95% confidence intervals in the linear regression analysis. **c,d**, The climatological mean value of OHC (solid lines) and the linear trend of the fractional area covered by high OHC (bars) ('Environmental factors') obtained from the ECMWF dataset in each month during the active season during 1981–2017 in the NH (**c**) and the SH (**d**). **e**–**h**, Same as in **a**–**d** but obtained from the HIS simulations. The climatological mean value of PI (solid lines) and the linear trend of the fractional area covered by high PI (bars) ('Environmental factors') in each month during the active season during 1981–2017 in the NH (**e**) and the SH (**f**). The climatological mean value of OHC (solid lines) and the linear trend of the fractional area covered by high OHC (bars) ('Environmental factors') in each month during the active season during 1981–2017 in the NH (**g**) and the SH (**h**). The error bars indicate the s.d. **i**–**l**, Same as in **a**–**d** but obtained from the GHG simulations. The climatological mean value of PI (solid lines) and the linear trend of the fractional area covered by high PI (bars) ('Environmental factors') in each month during the active season during 1981–2017 in the NH (**i**) and the SH (**j**). The climatological mean value of OHC (solid lines) and the linear trend of the fractional area covered by high OHC (bars) ('Environmental factors') in each month during the active season during 1981–2017 in the NH (**k**) and the SH (**l**). On the *x* axis, −1 indicates the previous year and 0 indicates the current year.

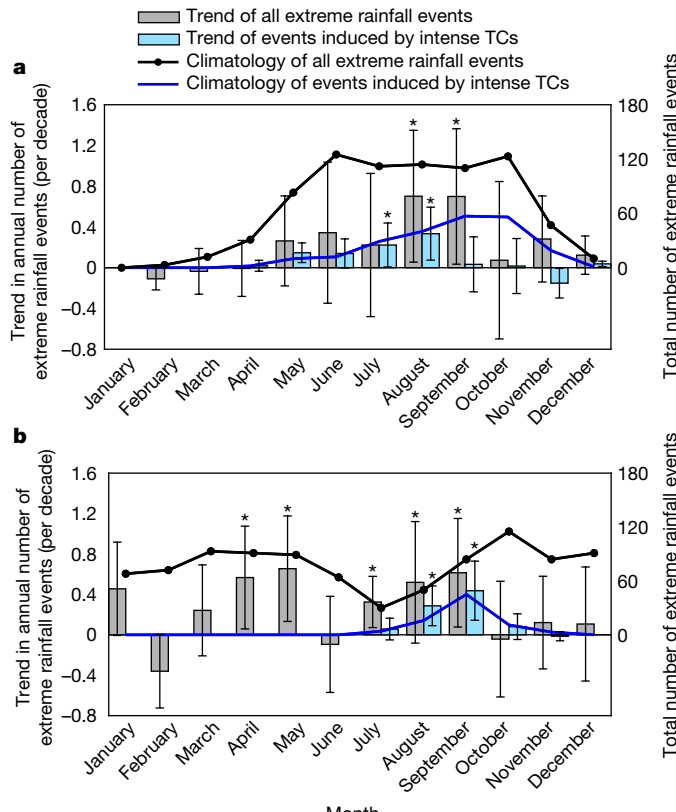

**Fig. 4 | The earlier seasonal occurrence of extreme rainfall produced by intense TCs. a,b,** The seasonal distribution of the total number of all extreme rainfall events (black lines), the linear trend in the annual number of all extreme rainfall events (grey bars; per decade), the total number of the extreme rainfall events induced by intense TCs (blue lines) and the linear trend in the annual number of the extreme rainfall events induced by intense TCs (blue bars; per decade) in each month from 1981 to 2017 in South China (105–120° E, 15–25° N; **a**) and the Gulf of Mexico (80–100° W, 25–35° N; **b**). The rainfall data are obtained from the Climate Prediction Center global unified gauge-based dataset and the intense TC data are obtained from the ADT-HURSAT dataset. *Significance at the 95% confidence level based on a non-parametric statistical test. The error bars indicate the 95% confidence intervals in the linear regression analysis.

to distinguish intense TCs from those that are less intense. We found that the oceanic conditions (for example, PI and OHC) become more beneficial for the intense TCs in the early season than those in the late season, thus favouring the seasonal advance of the intense TCs. In contrast, the changes in the atmospheric factors are either modest or insignificant, and thus, they do not contribute much to this earlier-shifting trend. The earlier onset of favourable oceanic conditions is detectable based on both the multimodel mean of the CMIP6 historical simulations and the CESM2 large ensemble. To investigate whether the detected earlier-shifting trend in favourable oceanic conditions is driven by anthropogenic or natural forcings, we compared the contributions of the GHG, natural and anthropogenic aerosol forcings based on individual forcing experiments from the DAMIP ('Model simulations', Fig. 3 and Extended Data Fig. 8). The earlier onset of favourable oceanic conditions is primarily driven by GHG forcing, whereas the contributions of natural and anthropogenic aerosol forcings are negligible or even negative. This earlier-shifting trend is projected to be amplified in the future (2080–2099) relative to the past (1981–2000) under a high-emission scenario ('Model simulations' and Extended Data Fig. 9a–d).

In South China and the Gulf of Mexico, the earlier onset of intense TCs contributes substantially to an earlier onset of extreme rainfall,

which could lead to the increased overlap period between the extreme rainfall produced by other weather systems (for example, the summer monsoon depression system) and that produced by intense TCs. This overlap could create devastating impacts that are well beyond any one of these events individually[33,34]. By drawing attention to the seasonal advance of intense TCs in this study, the need for further research and adaptation planning to protect those at high risk from TC-related damage is raised.

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

## Methods

### TC observations

The ADT-HURSAT[12] dataset provides globally homogenized TC records derived from satellite imagery during the period of 1981–2017. The ADT-HURSAT dataset is created with TC track data from the operational dataset and the TC intensity data estimated from global geostationary satellite imagery, which are homogenous in time and region. Global TC best-track data during the period of 1981–2021, obtained from the International Best Track Archive for Climate Stewardship v.4 (ref. 51), are exploited to further confirm the results based on the TC data derived from the ADT-HURSAT dataset. To analyse global trends, the ADT-HURSAT dataset is homogenized and used as a priority. There is some uncertainty in the absolute value estimates of TC intensity in the ADT-HURSAT dataset, and the best-track data are likely to have lower-intensity errors when looking at individual storms and seasons[52,53]. Consistent with the above findings obtained from the ADT-HURSAT dataset, the earlier-shifting trend of intense TC occurrence based on the best-track dataset is also statistically significant in both hemispheres (Table 1). Because the global trends obtained from the best-track data are more susceptible to data homogeneity problems, we include them as supporting evidence. The global TC data since 1981 are considered to be of high quality because geostationary satellites are routinely used in monitoring TCs, and most adjustments to their instrumentation and orbital positions were made in the 1970s.

Special attention is paid to intense TCs. An intense TC means its lifetime maximum intensity is over 110 kt, which is equivalent to a category 4–5 TC based on the Saffir–Simpson scale[8,17]. Among six basins where TCs occur, we consider only the western North Pacific, eastern North Pacific and North Atlantic in the NH as well as the South Indian Ocean and South Pacific in the SH. The North Indian Ocean is excluded because few intense TCs were observed in that basin. The overall TC events, which reach at least tropical storm intensity (over 35 kt) during their lifetime, are also considered.

### Occurrence time

The occurrence time of an intense TC event is defined as the date on which it first achieves its lifetime maximum intensity. This is determined by the relative change in TC intensity rather than the absolute value, which ensures the reliability of results[12,51]. Furthermore, measurements of TC intensity are often much more stable near the TC lifetime maximum intensity. Note that the TC season in the NH is defined as the current calendar year, whereas the TC season in the SH is defined from 1 July of the last calendar year to 30 June of the current calendar year as the seasonal distribution of TC activity in the SH straddles the calendar year.

The median value of intense TC occurrence dates during the active season is calculated. The active season is defined as June to November in the NH and December to April in the SH, comprising approximately 85% of all TC events. The early season is defined as the first half of the active season (that is, June to August in the NH and December to February in the SH), and the late season is defined as the second half of the active season (that is, September to November in the NH and March to April in the SH).

We consider the median value during the active season rather than the whole year to exclude the impact of sporadic TC events occurring in winter. To examine the extent to which this earlier-shifting trend may depend on how it is estimated, in addition to the median value during the active season used above (Fig. 2a,d), we also estimate this trend in terms of the expected occurrence of intense TCs based on a two-side bounded beta distribution assumption. The expected occurrence of intense TCs during the whole year is calculated by assuming that the seasonal cycle of intense TCs throughout the TC season approximately follows the beta distribution[54]. The expected value ($\mu = \alpha/(\alpha + \beta)$) of the occurrence date is calculated under the assumption that the seasonal cycle of intense TCs approximately follows a beta distribution, where $\alpha$ and $\beta$ are beta parameters. This assumption is justified because the seasonal occurrence date of intense TCs is bounded on both ends. Beta distribution parameters are usually fit using the method of the moment. Results from the expected occurrence of intense TCs support the same conclusion (Table 1).

### RI events

An RI event is identified if the TC intensity increases at least 35 kt over a period of 24 h, equivalent to the 95th percentile of the historical TC intensification rate[41]. Experiencing an RI process is considered as a reasonable criterion for distinguishing intense TCs from other TCs[36].

### Environmental factors

Four environmental factors that may be related to the RI process are considered, including PI, OHC, RH and VWS. PI and OHC are referred to as oceanic factors, whereas RH and VWS are atmospheric factors. PI is the theoretical maximum intensity that a TC can achieve and is determined by sea surface temperature and the associated thermodynamic profile[8,41]. We compute PI using monthly mean atmospheric temperature, specific humidity, sea-level pressure and sea surface temperature data[41], and the algorithm is available at ftp://texmex.mit.edu/pub/emanuel/TCMAX/. To represent the upper ocean thermal field possibly modulated by TCs, OHC (also called TC heat potential) is used and defined as the integrated heat content from the sea surface down to the 26 °C isotherm[55], $OHC = c_p \int_{D26}^{0} \rho(z)[T(z) - 26]dz$ (that is, where $c_p$ is the specific heat at constant pressure, $\rho(z)$ is the density of ocean water, $T(z)$ is the temperature of ocean water, $z$ is the vertical coordinate, $z = 0$ is the ocean surface and $z = D26$ is the vertical coordinate of the 26 °C isotherm). The data at 700 hPa were used for the RH. VWS, which suppresses the TC development process, is calculated as the magnitude of the vector difference between horizontal winds at 200 and 850 hPa.

The oceanic variables are obtained from the ECMWF[56] monthly Ocean Reanalysis database and the Global Ocean Data Assimilation System[57]. The atmospheric variables are obtained from the ERA-5 dataset[58], Japanese global atmospheric 55-year reanalysis[59] and Modern-Era Retrospective Analysis for Research and Applications v.2 (ref. 60). With improvements in assimilation schemes and numerical weather prediction models, the reanalysis reliability can vary considerably depending on the location and evaluation metrics[61].

The environmental thresholds for RI events are calculated following the method by ref. 43. Consistent with ref. 43, we found that stronger PI, higher OHC, higher RH and weaker VWS are all beneficial for RI. The exact value differs from that by Bhatia et al.[43], as we use 35 kt as the threshold of RI definition rather than the 30 kt used by Bhatia et al.[43]. This result suggests that it is reasonable to use the high quartiles of PI, OHC and RH and the low quartiles of VWS to indicate the environmental conditions that lead to RI. Although the local conditions experienced by TCs should be more relevant for the likelihood that they undergo RI, it cannot be obtained with good accuracy in the low-resolution climate model simulations in which the TC and RI process cannot be represented well (for example, historical and single-forcing experiments). Instead, the fractional area coverage by high-percentile PI and OHC, also corresponding well to RI events, is simple and can be directly calculated from the simulations. Here, the favourable conditions are quantified by the fractional area covered by high PI, high OHC, high RH and low VWS relative to the overall tropical ocean area (5–30° N, 5–30° S). The area covered by high PI, high OHC and high RH is defined as the sum of the area with the corresponding value above its 95th percentile, respectively, whereas the area covered by low VWS is defined as the sum of the area with the VWS lower than its 5th percentile. A sensitivity analysis was conducted where the 90th percentile of PI, OHC and RH and the 10th percentile of VWS were used. Consistently, an earlier shift of the seasonal cycle favourable oceanic conditions is obvious

(Extended Data Fig. 11a–d), whereas there is no significant change in the atmospheric conditions (Extended Data Fig. 11e–h).

### Model simulations

To detect the earlier-shifting trend of favourable oceanic conditions, we use the multimodel mean of environmental factors from phase 6 of the CMIP6 simulations[62]. The historical simulations during 1981–2014 and the future high-emission scenario (Shared Socioeconomic Pathway 585; SSP585) simulations during 2080–2099 of 20 models from CMIP6 are considered (Extended Data Table 1). The large ensemble from CESM2 (ref. 63) is also used, by which internal variability could be largely suppressed.

There are some biases in simulating ocean mesoscale conditions of the relatively low-resolution CMIP6 models, which may introduce uncertainty in the analysis of the TC-related environmental field[5,64,65]. Although the large-scale environmental fields are the focus of our study, it would be better to make a comparison of the results obtained from the high-resolution and low-resolution model outputs of CMIP6 HighResMIP simulations. The historical simulations (called hist-1950 in the HighResMIP) from these five pairs (a total of 10 models) of high-resolution and low-resolution models are used: CNRM-CM6-1-HR and CNRM-CM6-1; EC-Earth3P-HR and EC-Earth3P; ECMWF-IFS-HR and ECMWF-IFS-LR; HadGEM2-GC31-MM and HadGEM2-GC31-LL; and INM-CM5-H and INM-CM5-0. These model pairs contain all the variables needed. The HighResMIP multimodel ensemble analysis shows that the earlier onset of the high percentiles of PI and OHC is evident based on both the high-resolution and the low-resolution model outputs (Extended Data Fig. 9e–h). The consistency in the results based on the high-resolution and low-resolution outputs of CMIP6 HighResMIP simulations increases the confidence in our results based on the CMIP6 models.

To address whether the earlier onset of intense TCs is anthropogenic or natural forcing, all available experiments from the DAMIP[66] are used in this study. The DAMIP was designed to estimate the contributions of anthropogenic and natural forcings to observed global and regional climate changes. The experiments are conducted under individual forcings, including hist-GHG, in which only GHGs are prescribed; hist-nat, in which only natural forcings (solar activity and volcanic aerosols) are prescribed; and hist-aer, in which only anthropogenic aerosols are prescribed. In total, these nine models with all the outputs needed are used: ACCESS-CM2, BCC-CSM2-MR, CanESM5, CESM2, FGOALS-g3, GISS-E2-1-G, IPSL-CM6A-LR, MRI-ESM2-0 and NorESM2-LM. For each model, only one realization is used in each experiment.

### Extreme rainfall events

The occurrence frequency of an extreme rainfall event is defined as the number of days with a rainfall amount over 50 mm, which is approximately equivalent to the 95th percentile of the daily rainfall. The rainfall data are obtained from the Climate Prediction Center Global Unified Gauge-Based Analysis of Daily Precipitation dataset[67]. The extreme rainfall events are counted in South China (15–25° N, 105–120° E) and the Gulf of Mexico (25–35° N, 80–100° W).

The extreme rainfall events induced by intense TCs are calculated as a rainfall amount over 50 mm over intense TC-centred circles with a radius of 500 km. The radii of 500 km were chosen based on previous analyses of TC rainfall[68,69].

In addition, the persistent rainfall event is used as an example of compound hazards, which occur during the overlap period between extreme rainfall produced by other weather systems (for example, the summer monsoon depression system) and that produced by intense TCs. A persistent rainfall event can lead to disproportionate impacts by inducing a superimposed elevation of water levels and a shortage of disaster preparation resources[5,50,70]. Following a simple definition of the persistent rainfall event proposed by ref. 69,

the absolute threshold of daily precipitation amount exceeding 50 mm for at least three consecutive days is used. Note that a more elaborate analysis of the compound hazard of intense TCs and other high-impact events requires a careful definition of the compound hazard and quantitative examination of its change, which will be addressed in a future study.

### Statistical significance

The Mann–Kendall test is employed to determine if a statistically significant trend at a special level is achieved. This approach is non-parametric, so knowing the distribution for the data series is not required. This property enables the Mann–Kendall test to be applicable to the number of TC events or other extreme events in particular[71,72]. The Student's $t$-test is used to evaluate the 95% confidence intervals of the linear trend estimate[73,74]. The uncertainty of the multimodel mean is represented by the s.d. (ref. 75).

### Data availability

The data that support the findings of this study are all openly available online. In particular, the advanced Dvorak Technique–Hurricane Satellite data are available from the supporting information in ref. 11. The best-track data are available at https://www.ncei.noaa.gov/products/international-best-track-archive. The European Centre for Medium-Range Weather Forecasts monthly Ocean Reanalysis data are available at https://www.ecmwf.int/en/research/climate-reanalysis/ocean-reanalysis. The Global Ocean Data Assimilation System reanalysis data of ocean density and temperature are available at https://www.psl.noaa.gov/data/gridded/data.godas.html. The fifth-generation global atmospheric reanalysis data are available at https://cds.climate.copernicus.eu/#!/search?text=ERA5&type=dataset. The Japanese global atmospheric 55-year reanalysis data are available at https://climatedataguide.ucar.edu/climate-data/jra-55. The Modern-Era Retrospective Analysis for Research and Applications v.2 reanalysis data are available at https://disc.gsfc.nasa.gov/datasets/M2TMNXFLX_5.12.4/summary. The Coupled Model Intercomparison Project phase 6 model outputs are available at https://esgf-node.llnl.gov/search/cmip6/. The Community Earth System Model v.2 large ensemble dataset is available at https://www.cesm.ucar.edu/community-projects/lens2/data-sets. The rainfall data are available at https://psl.noaa.gov/data/gridded/data.cpc.globalprecip.html.

### Code availability

The analytical scripts are available. The codes used to calculate the observed seasonal advance of intense tropical cyclones, conduct the analysis of related factors and produce the main figures are available at https://doi.org/10.5281/zenodo.8163052. The potential intensity was calculated using the algorithm described in ref. 41.

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

**Acknowledgements** We thank the graduate students from the groups of X.Y. and F.S., including C. Wang, S. Zeng, J. Song, S. Lv and Y. Xiang, who helped to download and process the model simulation datasets. This research is jointly supported by the National Natural Science Foundation of China (Grant nos. 12102231, 11732008 and 42175029) and the Science and Technology Innovation Project of Laoshan Laboratory (Grant no. LSKJ202202201).

**Author contributions** K.S., X.Y. and F.S. designed the research. K.S. performed the analysis and drew all the figures. F.S. proposed the analysis of the CMIP6 model simulations and the seasonal cycle changes of extreme rainfall events. K.S. and F.S. wrote the first draft of the paper. All authors provided comments on different versions of the paper.

**Competing interests** The authors declare no competing interests.

**Additional information**
**Correspondence and requests for materials** should be addressed to Xiping Yu or Fengfei Song.

**a**

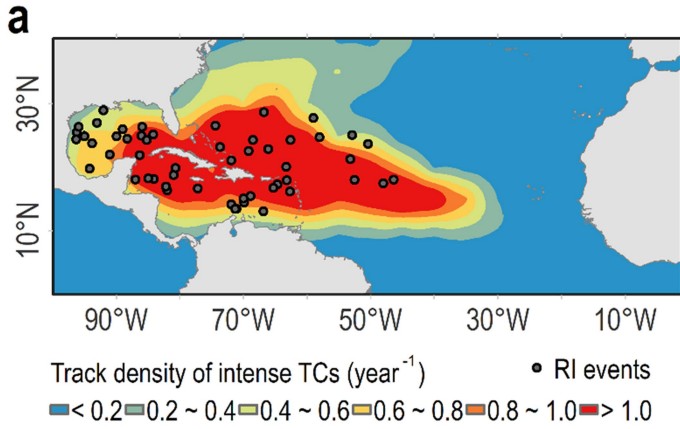

Track density of intense TCs (year⁻¹)    • RI events

< 0.2  0.2 ~ 0.4  0.4 ~ 0.6  0.6 ~ 0.8  0.8 ~ 1.0  > 1.0

**b**

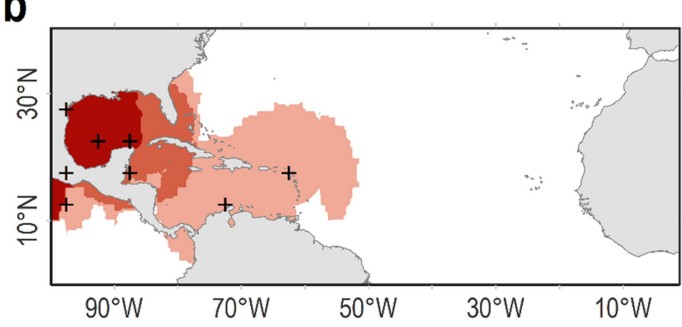

Trend of inter-seasonal difference (10⁻² decade⁻¹)

< -2.5    -1.5 ~ -0.5  0.5 ~ 1.5  > 2.5

-2.5 ~ -1.5  -0.5 ~ 0.5  1.5 ~ 2.5

**Extended Data Fig. 1 | Maps of the occurrence frequency of intense TCs and the inter-seasonal difference between the early and late seasons.** a, Distribution of the track density of intense TCs (shading) and the location of RI events (dots) over the North Atlantic basin during 1981–2017. b, Linear trends of the inter-seasonal difference in intense TC number between the early and late seasons (also part of Fig. 1c). The black crosses indicate significance at the 95% confidence level. The annual number of intense TCs is calculated over 5° × 5° boxes prior to computing trends. The basemaps in panels **a** and **b** were plotted using the Matplotlib basemap toolkit[76] with the geographical coordinate system World Geodetic system 1984 generated by the Global Positioning System (maintained by the National Oceanic and Atmospheric Administration).

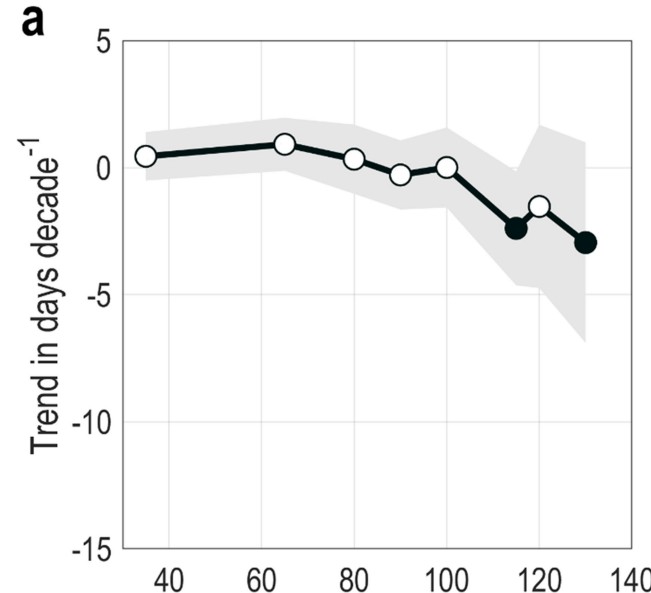

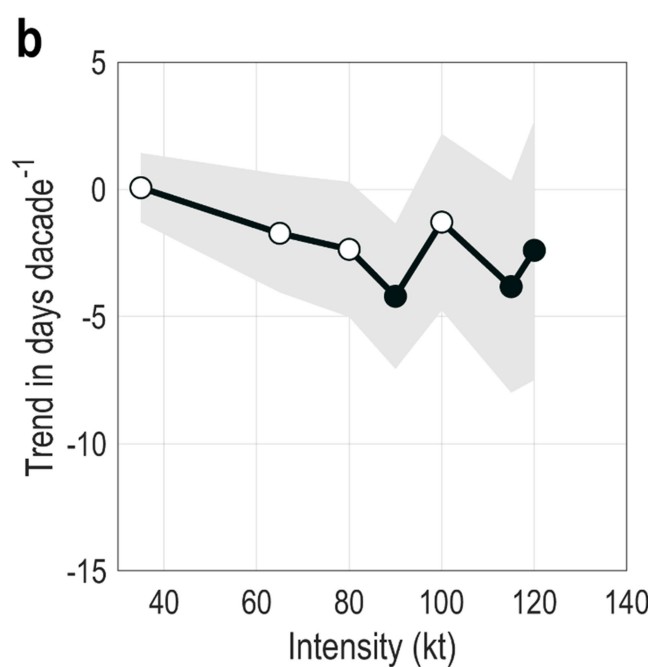

**Extended Data Fig. 2 | Shifting trends of the seasonal occurrence of TC events with different intensities.** Linear trends of the median value of the occurrence date (days per decade) of TC events with different intensities during the active season during 1981–2021 obtained from the best-track dataset in the (**a**) NH, and (**b**) SH. Shadings represent 95% confidence intervals in the linear regression analysis. Solid circles denote significant trends at the 95% confidence level based on a non-parametric statistical test, and open circles denote insignificant trends.

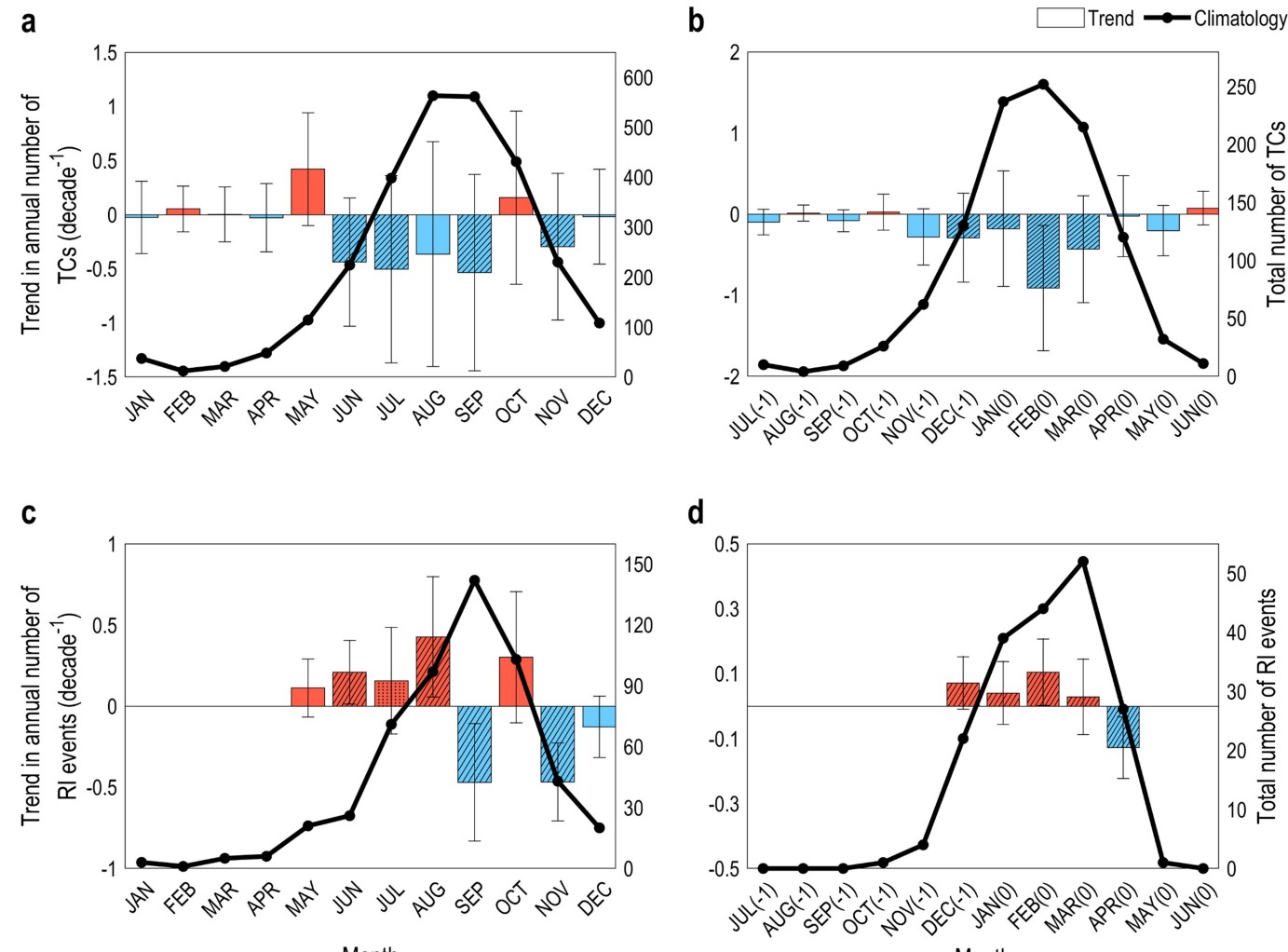

**Extended Data Fig. 3 | Observed changes in the seasonal cycle of TC occurrence and RI occurrence. a, b,** The seasonal distribution of the total number of TCs (solid line) and the linear trend in the annual number of TCs (bar; per decade) in each month from 1981 to 2017 obtained from the ADT-HURSAT dataset in the (**a**) NH, and (**b**) SH. The diagonal lines indicate significance at the 95% confidence level based on a non-parametric statistical test. The error bars indicate the 95% confidence intervals in the linear regression analysis. **c, d,** The seasonal distribution of the total number of RI events (solid line) and the linear trend in the annual number of RI events (bar; per decade) in each month from 1981 to 2017 obtained from the ADT-HURSAT dataset in the (**c**) NH, and (**d**) SH.

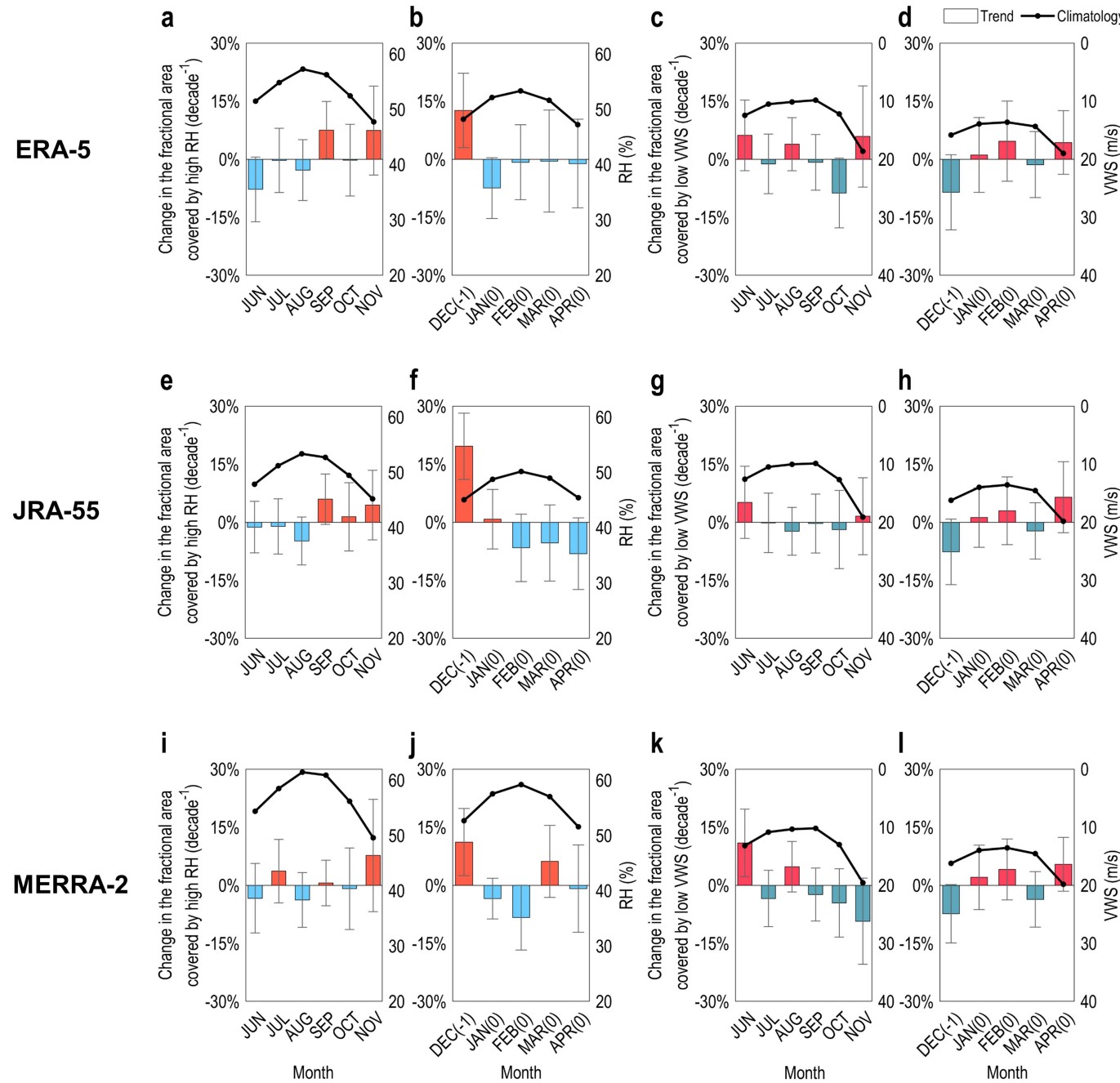

**Extended Data Fig. 4 | Observed changes in the seasonal cycle of atmospheric conditions based on different reanalyses.** For (up) with the atmospheric variables from the ERA-5, (middle) JRA-55, and (bottom) MERRA-2 datasets. **a**, **b**, The climatological mean value of relative humidity (RH; solid line) and the linear trend of the fractional area covered by high RH (bar; see Environmental factors of the Methods) in each month during the active season during 1981–2017 obtained from the ERA-5 reanalysis dataset in the (**a**) NH, and (**b**) SH. The error bars indicate the 95% confidence intervals in the linear regression analysis. **c**, **d**, The climatological mean value of vertical wind shear (VWS; solid line) and the linear trend of the fractional area covered by low VWS (bar; "Environmental factors") in each month during the active season during 1981–2017 obtained from the ERA-5 reanalysis dataset in the (**c**) NH, and (**d**) SH. **e**–**h**, Same as in **a**–**d**, respectively, but based on the JRA-55 dataset. **i**–**l**, Same as in **a**–**d**, respectively, but based on the MERRA-2 dataset.

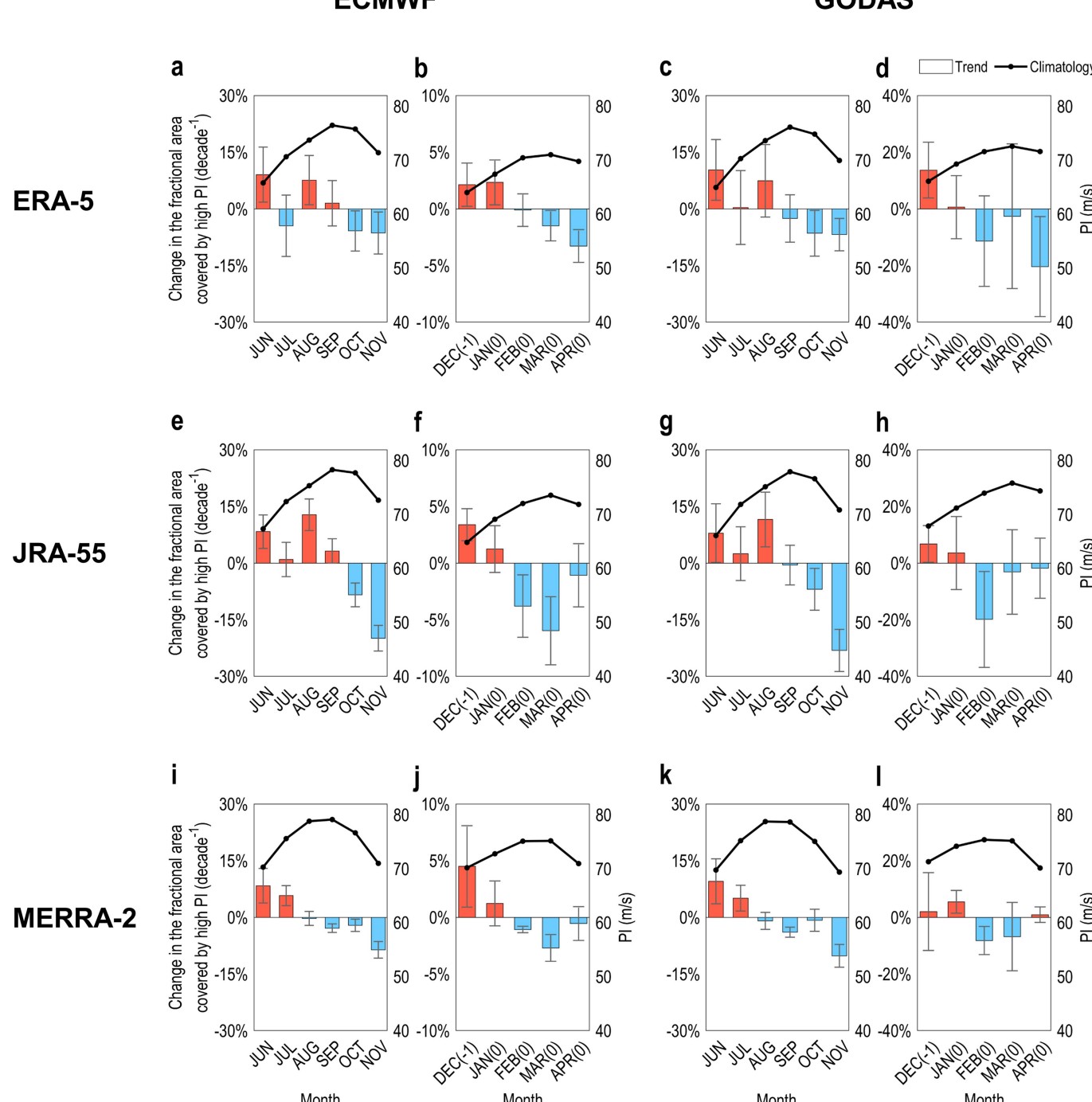

**Extended Data Fig. 5 | Changes in the fractional area covered by high PI based on different reanalyses.** For (left) with the oceanic variables from the ECMWF, and (right) GODAS datasets. For (up) with the atmospheric variables from the ERA-5, (middle) JRA-55, and (bottom) MERRA-2 datasets. **a**, **b**, The climatological value of PI (solid line) and the linear trend of the fractional area covered by high PI during the active season during 1981–2017 in the (**a**) NH, and (**b**) SH based on the oceanic variables from the ECMWF dataset and the atmospheric variables from the ERA-5 dataset. The error bars indicate the standard deviation. **c-d**, Same as in **a-b**, respectively, but based on the oceanic variables from the GODAS dataset. **e–h**, Same as in **a–d**, respectively, but based on the atmospheric variables from the JRA-55 dataset. **i–l**, Same as in **a–d**, respectively, but based on the atmospheric variables from the MERRA-2 dataset.

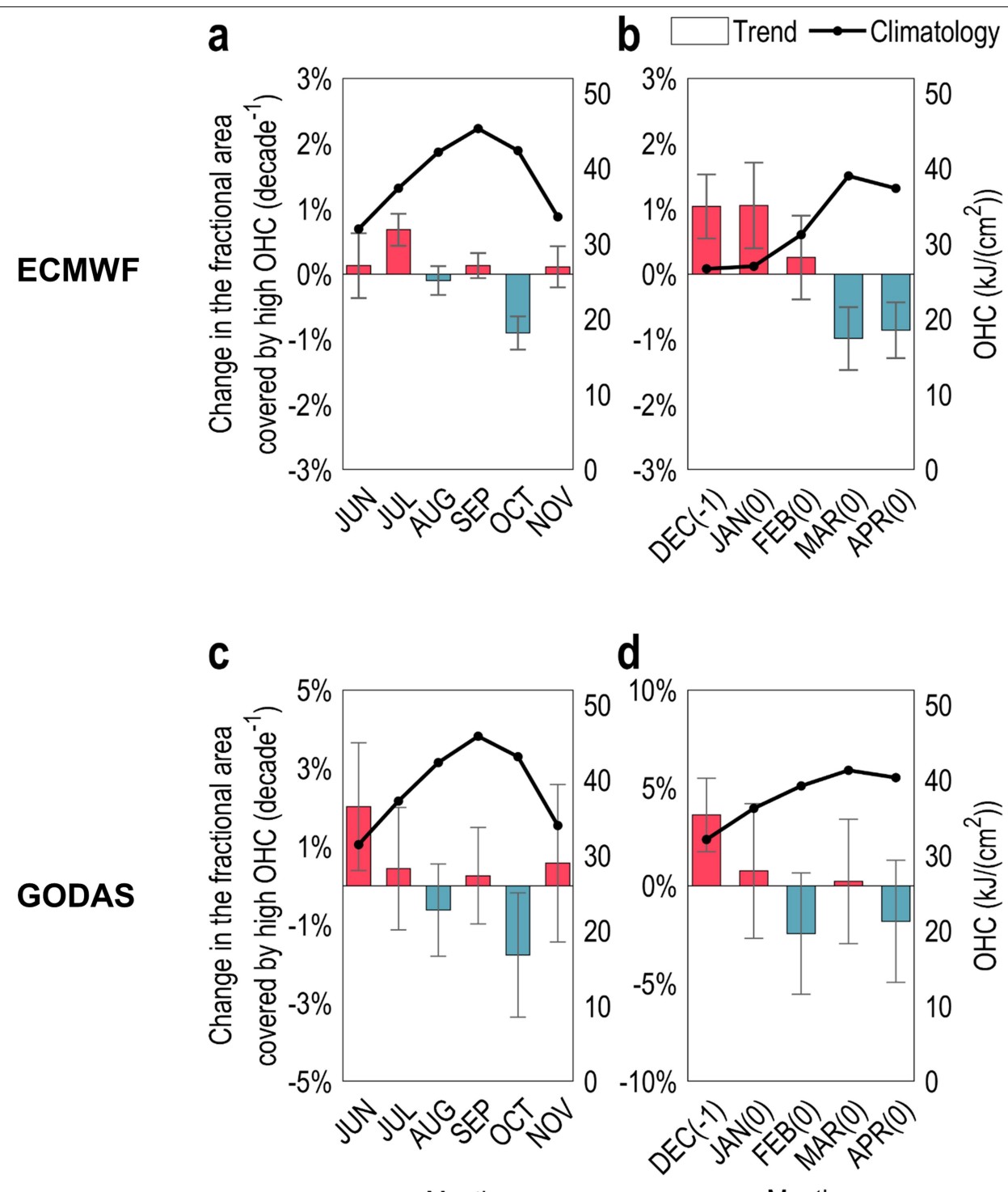

**Extended Data Fig. 6 | Changes in the fractional area covered by high OHC based on different reanalyses.** For (up) with the oceanic variables from the ECMWF dataset and (bottom) GODAS dataset. **a**, **b**, The climatological value of OHC (solid line) and the linear trend of the fractional area covered by high OHC during the active season during 1981–2017 in the (**a**) NH, and (**b**) SH based on the oceanic variables from the ECMWF dataset and the atmospheric variables from the ERA-5 dataset. The error bars indicate the standard deviation. **c-d**, Same as in **a**–**b**, respectively, but based on the oceanic variables from the GODAS dataset.

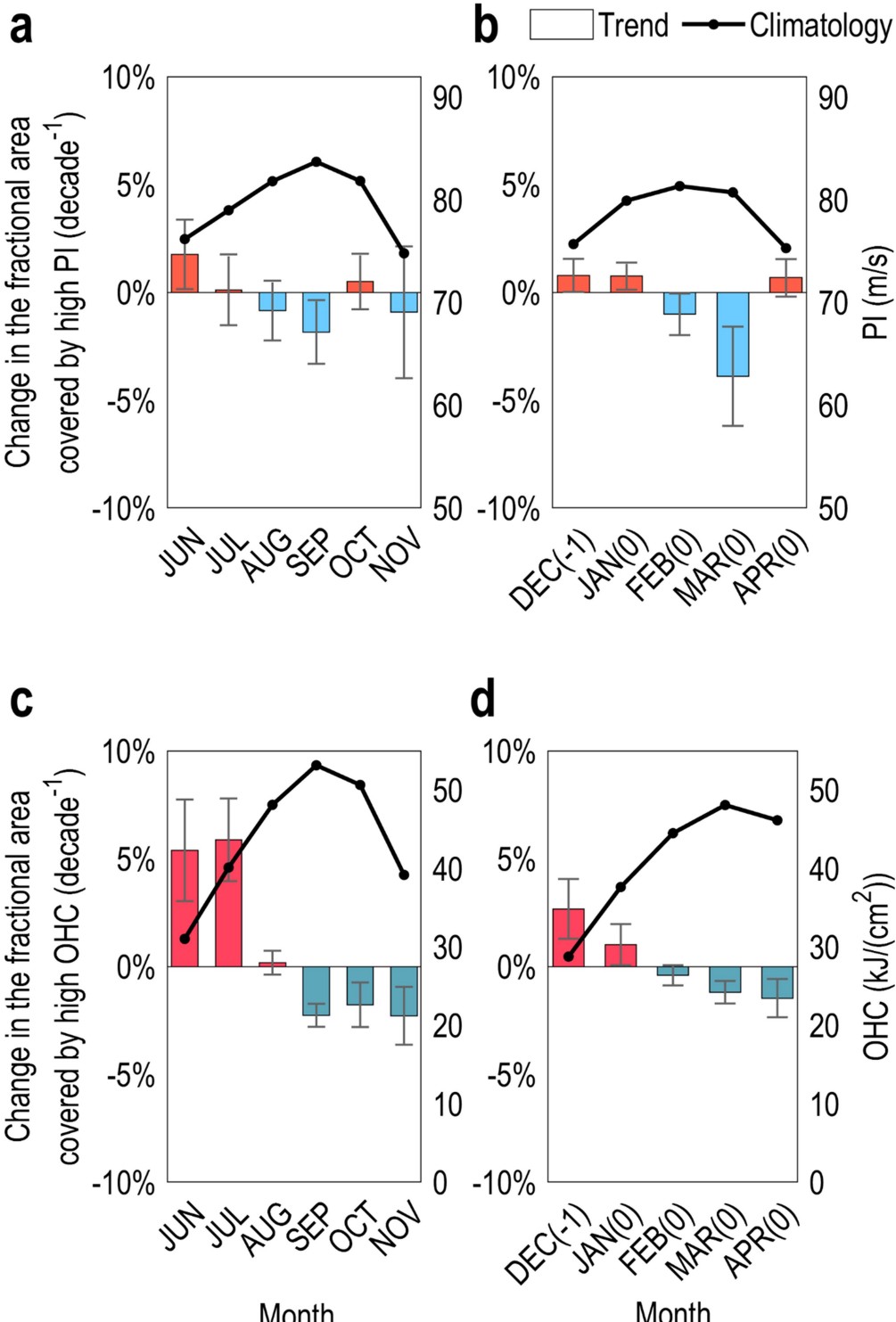

**Extended Data Fig. 7 | Simulated changes in the seasonal cycle of oceanic conditions obtained from the CESM2 large ensemble. a, b,** The multi-model mean of the climatological value of PI (solid line) and the linear trend of the fractional area covered by high PI (bar) in each month during the active season during 1981–2017 obtained from the CESM2 large ensemble in the **(a)** NH, and **(b)** SH. The error bars indicate the standard deviation. **c, d,** The multi-model mean of the climatological value of OHC (solid line) and the linear trend of the fractional area covered by high OHC (bar) in each month during the active season from 1981 to 2017 obtained from the CESM2 large ensemble in the **(c)** NH, and **(d)** SH.

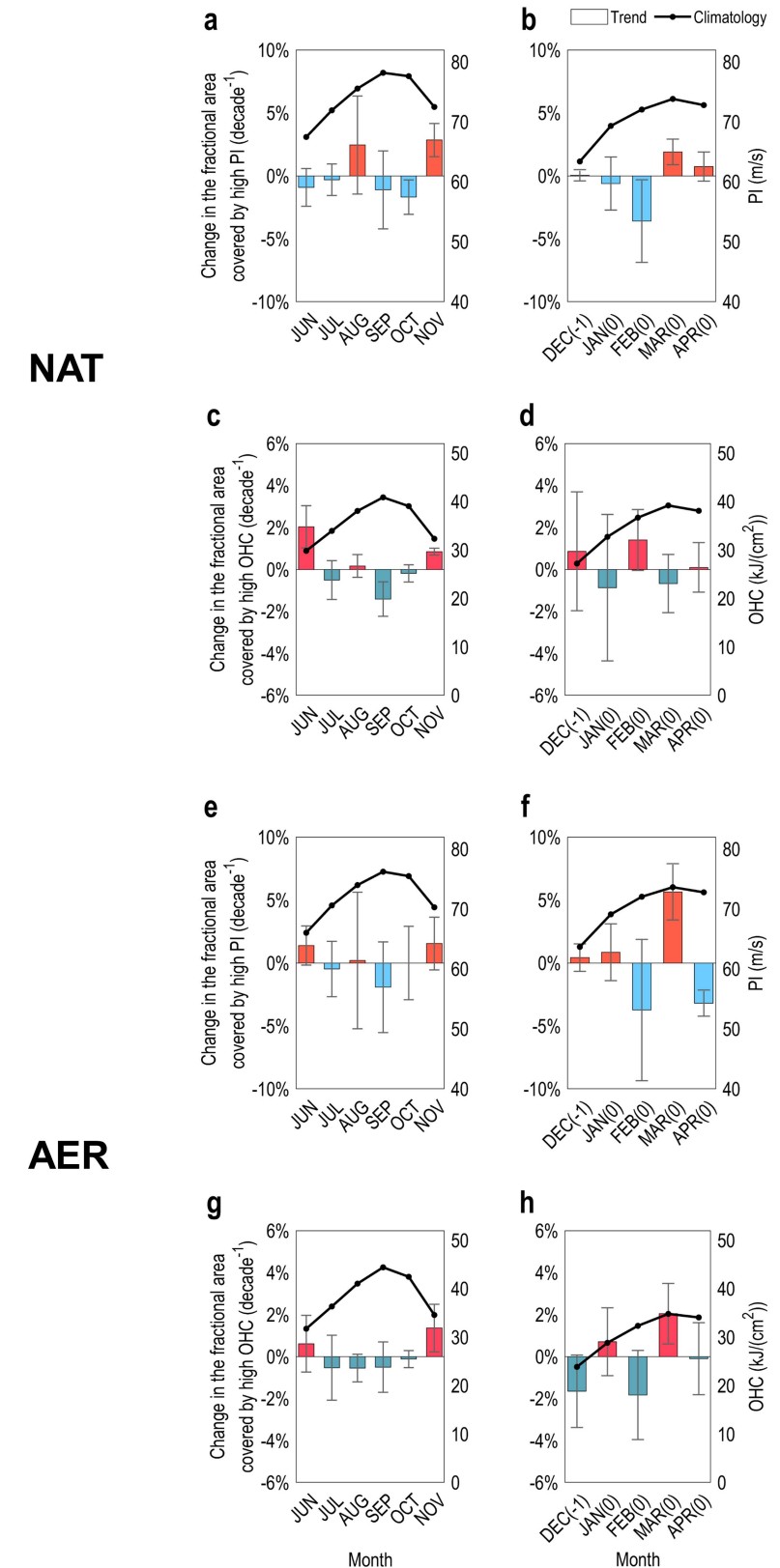

**Extended Data Fig. 8 | Changes in the seasonal cycle of oceanic conditions obtained from the DAMIP experiments under the natural (NAT) forcing only, and anthropogenic aerosol (AER) forcing only.** For (up) the multi-model mean of experiments forced by the NAT, and (bottom) experiments forced by the AER. **a, b**, The climatological value of PI (solid line) and the linear trend of the fractional area covered by high PI (bar) in each month obtained from the NAT experiments during the active season during 1981–2017 in the (**a**) NH, and (**b**) SH. The error bars indicate the standard deviation. **c, d**, The climatological value of OHC (solid line) and the linear trend of the fractional area covered by high OHC (bar) in each month obtained from the NAT experiments during the active season during 1981–2017 in the (**c**) NH, and (**d**) SH. **e–h**, Same as in **a–d**, respectively, but obtained from the AER experiments.

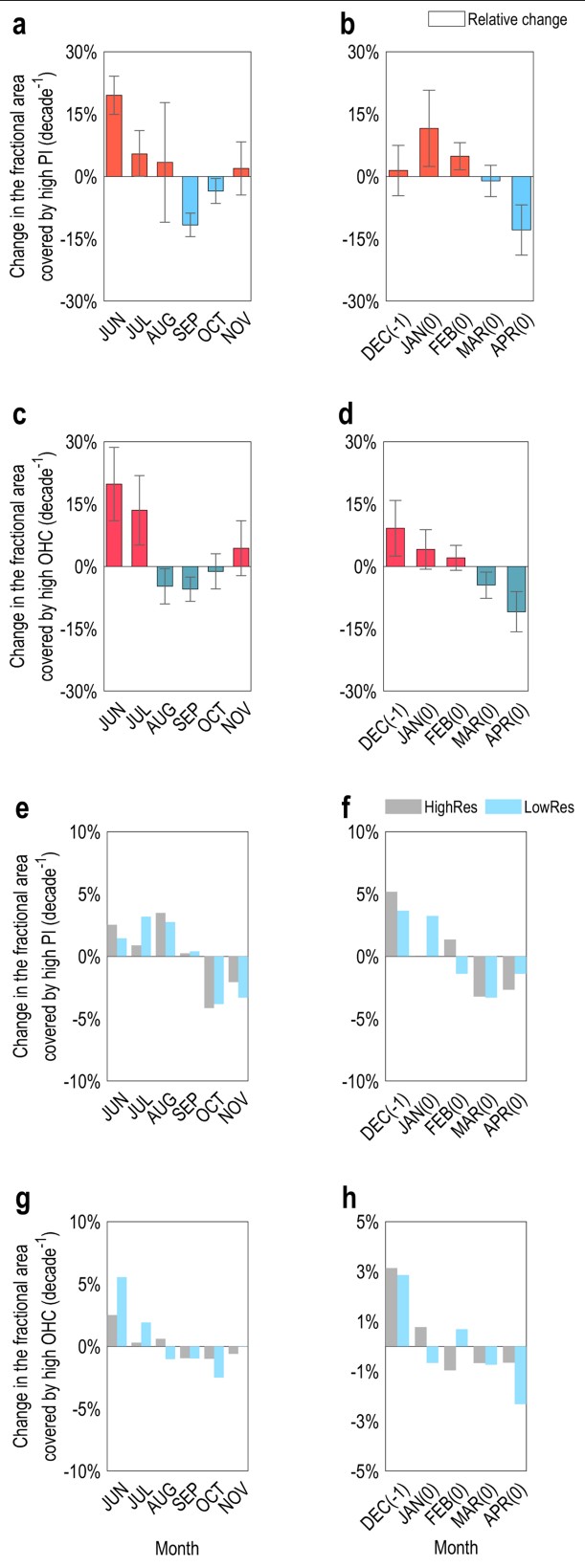

**Extended Data Fig. 9 | Simulated changes in the seasonal cycle of oceanic conditions obtained from the CMIP6 simulations under the high-emission scenario SSP585 and HighResMIP simulations.** For (top) the relative change between the period of 2080–2099 under the high-emission scenario SSP585 and the period of 1981–2000 based on the multi-model mean of CMIP6 simulations, and (bottom) HighResMIP simulations. **a**, **b**, The relative change of the fractional area covered by high PI (bar) in the active season between the period of 2080–2099 under the high-emission scenario SSP585 and the period of 1981–2000 in the (**a**) NH, and (**b**) SH based on the multi-model mean of CMIP6 simulations. The error bars indicate the standard deviation. **c**, **d**, The relative change of the fractional area covered by high OHC (bar) in the active season between the period of 2080–2099 under the high-emission scenario SSP585 and the period of 1981–2000 in the (**c**) NH, and (**d**) SH. **e**, **f**, The linear trend of the fractional area covered by high PI in each month during the active season during 1981–2017 obtained from the high resolution (HighRes; grey bars) outputs and the low resolution (LowRes; blue bars) outputs in the (**e**) NH, and (**f**) SH. **g**, **h**, Same as in **e**, **f**, respectively, but for the linear trend of the fractional area covered by high OHC.

**a**

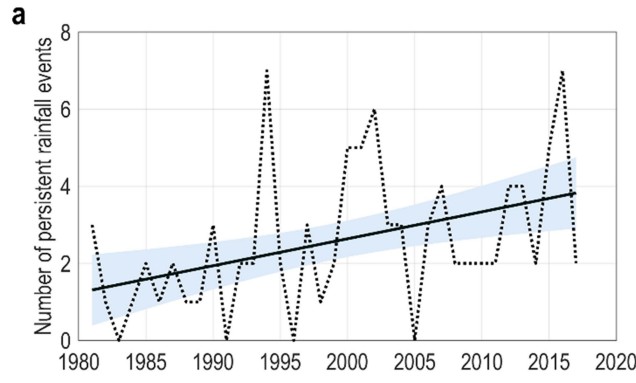

**b**

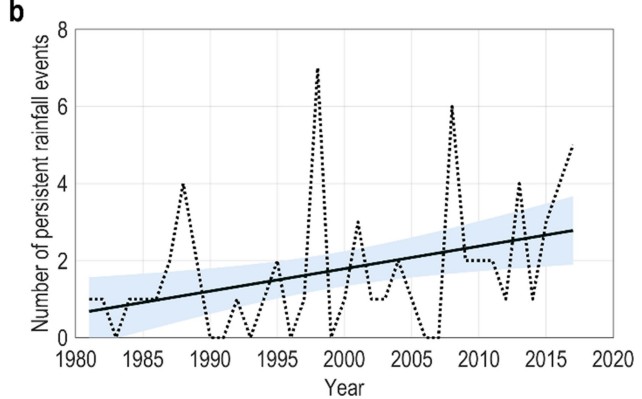

**Extended Data Fig. 10 | Changes of summer persistent rainfall frequency in the observation. a, b,** Time series of the annual number of the persistent rainfall events (dashed line) and its trend (solid line) in (**a**) South China, and (**b**) the Gulf of Mexico during July-September during 1981–2017. The shadings represent 95% confidence intervals in the linear regression analysis.

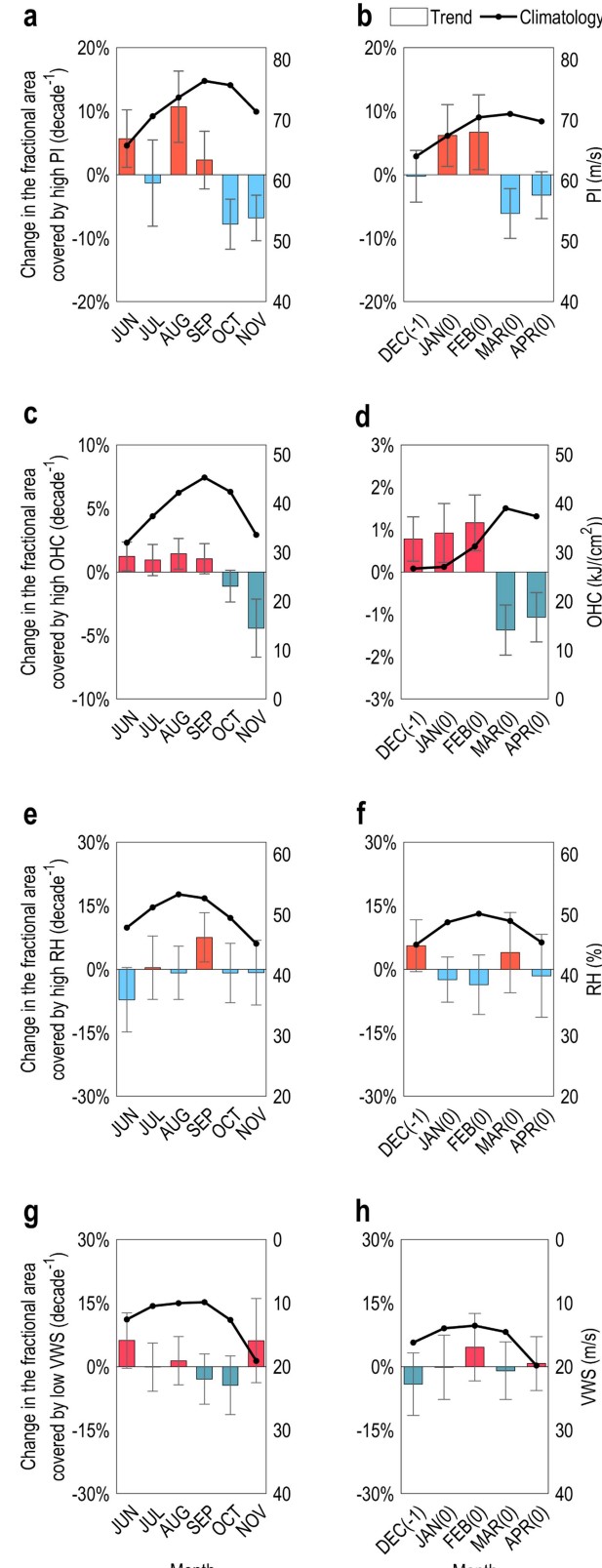

**Extended Data Fig. 11 | Sensitivity analysis of the thresholds of environmental factors. a, b**, The climatological mean value of PI (solid line) and the linear trend of the fractional area covered by the PI value above its 90th percentile (bar) in each month during the active season during 1981–2017 in the (**a**) NH, and (**b**) SH. The calculation of PI is based on the oceanic variables from the ECMWF dataset and the atmospheric variables from the ERA-5 dataset. The error bars represent 95% confidence intervals in the linear regression analysis. **c, d**, The climatological mean value of OHC (solid line) and the linear trend of the fractional area covered by the OHC value above its 90th percentile (bar) obtained from the ECMWF dataset in each month during the active season during 1981–2017 in the (**c**) NH, and (**d**) SH. **e, f**, The climatological mean value of RH (solid line) and the linear trend of the fractional area covered by the RH value above its 90th percentile (bar) obtained from the ERA-5 reanalysis dataset in each month during the active season during 1981–2017 in the (**e**) NH, and (**f**) SH. **g, h**, The climatological mean value of VWS (solid line) and the linear trend of the fractional area covered by the VWS value below its 10th percentile (bar) obtained from the ERA-5 reanalysis dataset in each month during the active season during 1981–2017 in the (**g**) NH, and (**h**) SH.

**Extended Data Table 1 | The CMIP6 models, including ocean and atmosphere models, their resolutions and references**

| No. | Model | Ocean Res. | Atmos. Res. | Reference |
|---|---|---|---|---|
| 1 | ACCESS-CM2 | MOM5 (1°) | UKMO UM (1.875°x1.25°) | Bi et al. (2020) |
| 2 | BCC-ESM1 | MOM4 (1°) | BCC AGCM3-Chem (2.8°) | Wu et al. (2020) |
| 3 | CAMS-CSM1-0 | MOM4 (1°) | ECHAM5 (1°) | Rong et al. (2019) |
| 4 | CanESM5 | CanNEMO (1°) | CanAM5 (2.8°) | Swart et al. (2019) |
| 5 | CAS-ESM2-0 | LICOM2 (1°) | IAP AGCM5 (1.4°) | Jin et al. (2021) |
| 6 | CESM2 | POP2 (1°) | CAM6 (1°) | Danabasoglu et al. (2020) |
| 7 | CIESM | POP2 (1°) | CAM5 (1°) | Lin et al. (2020) |
| 8 | CMCC-ESM2 | ORCA (1°) | CAM5.3 (1°) | Lovato et al. (2022) |
| 9 | E3SM-1-1 | MPAS-Ocean (0.5°) | EAMv1 (1°) | Zheng et al. (2022) |
| 10 | FGOALS-g3 | LICOM3 (0.5°) | GAMIL3 (2°) | Li et al. (2020) |
| 11 | FIO-ESM-2-0 | POP2 (1°) | CAM5 (1°) | Bao et al. (2020) |
| 12 | GFDL-ESM4 | MOM6 (0.5°) | AM4 (1°) | Dunne et al. (2020) |
| 13 | GISS-E2-1-G | GISS Ocean v1 (1°x1.25°) | GISS-E2.1 (2°x2.5°) | Kelly et al. (2020) |
| 14 | INM-CM5-0 | INM-OM5 (0.5°x0.25°) | INM-AM5-0 (2°x1.5°) | Volodin and Gritsun (2018) |
| 15 | IPSL-CM6A-LR | NEMO (1°) | LMDZ (2.5°x1.3°) | Boucher et al. (2020) |
| 16 | MCM-UA-1-0 | MOM1 (1.875°x2°) | R30L14 (3.75°x2.5°) | Stouffer (2019) |
| 17 | MPI-ESM1-2-LR | MPIOM1.6 (1.5°) | ECHAM6.3 (2°) | Mauritsen et al. (2019) |
| 18 | MRI-ESM2-0 | MRI.COM4.4(1°) | MRI-AGCM3.5 (1°) | Yukimoto et al. (2019) |
| 19 | NESM3 | NEMO v3.4 (1°) | ECHAM6.3 (2.5°) | Cao et al. (2018) |
| 20 | NorESM2-LM | MICOM (1°) | CAM-OSLO (2°) | Seland et al. (2020) |

Refs. 77–96.