## [Peer Review File · Nature]

Manuscript Title: Seasonal Advance of Intense Tropical Cyclones in A Warming Climate

Redactions – unpublished data

Reviewer Comments & Author Rebuttals

Reviewer Reports on the Initial Version:

Referees' comments:

Referee #1 (Remarks to the Author):

Review of Seasonal Advance of Intense Tropical Cyclones in A Warming Climate by Shan et al.

This article investigates seasonal shifts in intense tropical cyclones (TCs) (> 110 kts) and finds that the occurrence of intense TCs is shifting earlier in the TC season. Statistically significant trends are found both globally and in individual basins. This work adds to the body of literature that has detected changes in TC activity and is of significant interest to the broader community given the societal impacts of TCs. To my knowledge, this particular observed trend has not been previously identified so the results are novel.

In association with the trend towards earlier occurrence of the most intense TCs, the median value of the RI occurrence data has also been shifting earlier, consistent with the expectation that most intense TCs undergo RI. This is related to changes in thermodynamic environments, in which high percentiles of potential intensity and ocean heat content are also shifting towards earlier occurrence dates. These shifts in the timing of favorable environments for intensification are also seen in historical CMIP6 simulations, which leads the authors to conclude that anthropogenic warming has caused the trends.

The authors also relate the trends in the seasonal occurrence of intense TCs to trends in extreme rainfall events. They note that, at least in South China, there is historically a local minima in extreme rainfall events in mid-summer, but that this time period has seen the largest increasing trend in extreme rainfall events. They attribute this to an increasing trend in extreme rainfall produced by intense TCs during that time period due to the shift towards having intense TCs earlier in the season.

The methods are generally sound, with appropriate use of statistics and quantification of uncertainty. The paper is well-written. However, while I think this paper is worthy of publication, there are some revisions that should be made before it can be published. Some of the conclusions seem a bit overstated and some methods need to be clarified.

1. The most closely related study is that of Truchelut et al 2022 which found a trend towards earlier onset of the TC season in the North Atlantic in association with changes in environmental conditions. However in this study, when all TC events are considered (extended data Figure 1), no shifts towards earlier in the season are identified. There are a variety of different factors that could explain this - looking at the whole Northern Hemisphere versus just the North Atlantic, computing the trends in

number of TC events on a monthly basis (I think)? versus finer temporal resolution - but the authors should explicitly relate their results to the Truchelut study and discuss the differences.

2. Line 261, Lines 218-221: The conclusion that changes in oceanic conditions are mainly caused by external forcing is not as well-supported as it could be. This seems to be based on the fact that the trends towards earlier onset of high percentile PI and OHC are seen also in the CMIP6 historical simulations and in future projections. But this does not directly imply attribution! Are the authors arguing that the CMIP6 multi-model mean averages out internal variability and reveals the forced response, and so since we see similar trends to observations, that indicates the observed trends are due to the forced response? If so, that needs to be made clear (for example, is it the multi-model mean that is being plotted?). A stronger attribution statement could be made by using a large ensemble (so that internal variability could be averaged out) in combination with historical versus natural-only simulations.

3. Lines 262-264, Lines 241-243, lines 61-65: The earlier seasonal occurrence of extreme rainfall produced by intense TCs is used to argue that the earlier seasonal occurrence of intense TCs increases the risk of compound disasters by causing more overlap between intense TCs and other extreme events. However, this conclusion seems like a bit of a reach. If the intense TCs are the event causing the changes in extreme rainfall, then that is not necessarily a compound disaster. Just because the shift in intense TCs might be “merging the double peaks in the seasonal cycle of extreme rainfall events” does not mean there is a compound event, per se. Changes in compound extremes are certainly highly impactful and important to understand, but doing so requires a careful definition of the compound event and an examination of changes in its occurrence specifically, rather than just a general examination of changes in the seasonal distribution of extreme rainfall and of intense TCs. Instead, the authors could state that the earlier onset of intense TCs also leads to an earlier onset in extreme rainfall associated with them, and this could contribute to extreme rainfall occurring during unexpected times of year, or something like that.

4. Lines 239-241: An example for South China is given. Is it generalizable to other regions? Be careful about making broad conclusions based on one specific region.

5. Lines 194-196, Lines 331-335: It is not clear exactly how the trends in the occurrence of the favorable environmental conditions are being computed. Is the insinuation that if a larger spatial area has PI greater than the 95th percentile relative to the whole tropics, that more RI events will occur? It is not clear. Also, why should the 95th percentile of PI or OHC necessarily correspond to the 95th percentile of TC intensity changes? This choice should be justified by a more careful analysis of the environmental conditions that lead to RI, as performed by (for example) Bhatia et al 2022.

6. Lines 38-39: While the cited study (Chand et al 2022) does suggest a declining number of TCs over the past century (by explicitly tracking TCs in 20th century reanalysis), other methodologies show different results (Emanuel, K., 2021: Atlantic tropical cyclones downscaled from climate reanalyses show increasing activity over past 150 years. *Nature Communications*, 12, 7027, <https://doi.org/10.1038/s41467-021-27364-8>), which should be acknowledged.

7. Figure 1 and other similar figures: What exactly is plotted on the line and the y-axis? Is it the total number of events in that month across the entire record? The numbers are too high to represent the average number of events in that month per year. Please clarify.

8. Figure 1c: How are these trends calculated spatially? Is the intraseasonal difference in intense TC number computed for each grid point? And if so, what grid is used?

9. Figure 2 b,e: I know statistical significance is indicated by the symbol type, but could error bars be added to these plots?

10. Figure 2 c,f: Is it also the median value of RI occurrence date that is plotted?

11. Line 117: Suggest specifying open circles correspond to the insignificant trends

12. Line 319-323: Why are these variables defined? They are not shown in any equation anywhere.

13. Regarding the model simulations, what is being plotted in Figure 3? The multi-model mean?

14. Regarding the extreme rainfall events (above 50mm, which is stated to be the 95th percentile of daily rainfall) plotted for South China in Figure 4, the methods should indicate the specific region that is being considered.
15. Extended data Figure 1: Should the y-axis label be TC number, rather than Intense TC number? The caption indicates that all TC events are being considered.
16. Extended data Figure 1 and Extended data Figure 2: Please provide a full description of the plot in the caption, rather than saying it is the same as Figure 1a,b.
17. Extended data Figure 3: The caption refers to RI events in panels a,b as being plotted with a red line, but the plot shows a pink/purple line.
18. Line 329-330, In extended data Figure 6, the seasonal trends in the occurrence of high relative humidity and low vertical wind shear are compared in two different reanalysis datasets – ERA-5 and the NCEP/NCAR reanalysis. It is good that multiple reanalyses are considered, but the NCEP/NCAR reanalysis is somewhat outdated (and has known biases some fields that influence TCs, such as upper tropospheric/lower stratospheric temperatures), so it would be beneficial to compare to a newer reanalysis such as JRA-55, MERRA-2, or CFSR. In that case, the trends in potential intensity could also be compared in a second reanalysis.
19. Extended data Figure 7: For clarity, the caption should indicate which future scenario the CMIP6 simulations are using to arrive at the 2080-2099 period.

Referee #2 (Remarks to the Author):

Recommendation

This is an interesting study of seasonality change in intense tropical cyclones, with potentially important implications for understanding compound hazards in populous regions, a key component of near-future climate risk. In my view, the manuscript is generally suited to the readership of Nature, focussing on a sufficiently important topic, but is not publishable in its current form. The authors have identified consistent seasonal changes in area-mean oceanic environmental quantities and tropical cyclones, which is an important and novel observation, although other studies have examined similar environmental metrics in relation to TCs. However, the current manuscript falls short of definitively linking oceanic changes with tropical cyclone activity. The introduction refers to detection and attribution of shifts in TC behaviour to climate change, but the paper does not present any such analysis. Additionally, the North Atlantic, where no seasonal changes are identified in the authors' analysis, is largely not discussed. These issues are explained below, and the authors should consider addressing these and resubmitting to Nature or, perhaps more appropriately, Nature Climate Change.

Major comments

1. **Oceanic vs atmospheric factors in explaining earlier-shifting trends.** The authors logically look to explain the earlier-shifting trends in intense TCs / RI events by analysing oceanic environmental factors (PI and OHC) and atmospheric factors (wind shear and humidity). The authors justify focussing on oceanic factors by appealing to their shared seasonality with intense TCs, and this analysis is strengthened by the inclusion of two oceanic datasets. However, the atmospheric factors are not as well explained. Firstly, the reanalysis and models are not in agreement and, secondly, some significant trends are contrary to the oceanic results. Changes in wind shear across basins can be induced by global teleconnections, which in turn modulate seasonal TC activity, so the authors should not ignore the atmospheric factors.

2. **Model shortcomings.** The authors should be more critical of the relatively low-resolution CMIP6 models and their current shortcomings in analysis of TC environments in models. The ocean mesoscale, which can be important for TC intensification, is not represented in CMIP6 models. While I understand that the authors focus on large-scale environmental fields, which are represented reasonably in some models, there are improvements in sea-surface temperature biases and large-scale atmospheric circulation in higher-resolution

models, such as HighResMIP, which the authors should consider exploiting to strengthen their analysis.

3. **Clarifications and improvements to the text.** In several places, phrasing and language are vague and the text requires revision using appropriate and concrete language (frequency, particular hazards etc). See line-by-line comments for further detail. Additionally, author names in references are not formatted consistently.

Line-by-line comments

Line 21: “globally”—not sure this is entirely true. The North Atlantic is an important exception.

Lines 25–26: There could be multiple factors for the RI trend, so revise this statement. Lines 40–42: Other evidence suggests some poleward shift may be due to changes in seasonality (Feng et al., 2021)—this should be briefly included in the text.

Line 48: “serious consequences” is a bit vague. Be more specific and indicate particular hazards.

Line 51: By “other” do you mean less intense?

Line 54: “TCs are more concentrated in summer” is vague. Are you referring to frequency here? And is this statement true across basins? The seasonal peak is often not during JJA (for the NH).

Line 57: I don’t think this is needed.

Lines 61–65: A statement is needed at the start of this paragraph explaining that the paragraph discusses seasonality of different type of high-impact weather and the notion of compound risk.

Lines 86–87: It’s interesting that there is no change in the North Atlantic, only in the Gulf of Mexico. PI has increased in this basin (Klotzbach et al., 2022). The authors will need to discuss this further.

Lines 132–137: These results (Fig. 2b,e) are very interesting, but the authors should show the curves for both ADT-HURSAT and best-track data. The authors are correct that ADT-HURSAT is more appropriate for trend analysis, but they should acknowledge that ADT-HURSAT intensities may not be as accurate as the best-track data for the most intense TCs. In other words, ADT-HURSAT sacrifices intensity accuracy for homogenisation for trend analysis.

Lines 166–173: Bhatia et al. (2022), who show changes in global trends in RI, PI and TC environments based on the same dataset, is cited in the manuscript. This paper also discusses area-mean versus local TC environments, and some discussion of the two approaches would be a useful addition to the present manuscript. Reading through this submission, I think that some analysis of local TC environments (i.e., within a radius of TC centres) would be a worthwhile addition, if only in observations.

Line 203: Ext. Data Table 2 does not provide more information—see comment on tables below.

Lines 205–208: Explain the large October decrease in panel e.

Line 262. “will be further amplified under global warming”—this is likely true, but the authors have not shown, for example, that in control simulations without anthropogenic factors, seasonality shifts do not occur. Analysis of control simulations would be a necessary basis on which to make attribution-type statements.

Line 341: Change to Ext. Data Table 2.

Comments on figures and tables

Fig 1c. The cross symbols are quite coarse. Please replace with finer-scale hatching to show exactly where is significant more clearly.

Fig. 2. Show the trend uncertainty in panels a, c, d, f.

Ext. Data Table 2. This does not offer much information. Please add at least these details about each model: ocean and atmosphere resolution, ocean model, mean TC activity, references.

References

Bhatia, K., A. Baker, W. Yang, G. Vecchi, T. Knutson, H. Murakami, J. Kossin, K. Hodges, K. Dixon, B. Bronselaer, and C. Whitlock, 2022: A potential explanation for the global increase in tropical cyclone rapid intensification. *Nature Communications* 13, 6626

Feng, X., N. P. Klingaman, and K. I. Hodges, 2021: Poleward migration of western North Pacific tropical cyclones related to changes in cyclone seasonality. *Nature Communications* 12, 6210

Klotzbach, P. J., K. M. Wood, C. J. Schreck III, S. G. Bowen, C. M. Patricola, and M. M. Bell, 2022: Trends in Global Tropical Cyclone Activity: 1990–2021. *Geophysical Research Letters* 49, e2021GL095774

Author Rebuttals to Initial Comments:

Author Response to Referee#1 of “Seasonal Advance of Intense Tropical Cyclones in A Warming Climate”

General Comments

This article investigates seasonal shifts in intense tropical cyclones (TCs) (> 110 kts) and finds that the occurrence of intense TCs is shifting earlier in the TC season. Statistically significant trends are found both globally and in individual basins. This work adds to the body of literature that has detected changes in TC activity and is of significant interest to the broader community given the societal impacts of TCs. To my knowledge, this particular observed trend has not been previously identified so the results are novel.

Response

Thanks for the encouragement. We appreciate the insight and thoughtfulness of the referee’s comments and suggestions and will attempt to address each of them fully here.

In association with the trend towards earlier occurrence of the most intense TCs, the median value of the RI occurrence data has also been shifting earlier, consistent with the expectation that most intense TCs undergo RI. This is related to changes in thermodynamic environments, in which high percentiles of potential intensity and ocean heat content are also shifting towards earlier occurrence dates. These shifts in the timing of favorable environments for intensification are also seen in historical CMIP6 simulations, which leads the authors to conclude that anthropogenic warming has caused the trends.

Response

We thank the referee for the comment. We originally estimated the seasonal changes of the high percentiles of oceanic factors (potential intensity and ocean heat content) based on the CMIP6 multi-model mean. Following the referee’s suggestion,

we have performed this analysis based on the Community Earth System Model Version 2 (CESM2) large ensemble, which contains 100 members. The internal variability could be largely suppressed when we take the ensemble mean. It is found that the high percentiles of oceanic factors show significant trends towards earlier onset (please see below), in agreement with that based on the multi-model mean of the CMIP6 historical simulations. Further, to reveal the relative contributions of different forcings, we have estimated the seasonal changes of the high percentiles of oceanic factors based on the Detection and Attribution Model Intercomparison Project (DAMIP) experiments under the greenhouse gas forcing only, natural forcing only, and anthropogenic aerosol forcing only, respectively. Results (see below) suggest that the earlier onset of favorable oceanic conditions is primarily driven by the greenhouse gas forcing, while the contributions of the natural and anthropogenic aerosol forcings are negligible or even negative.

We think the additional analyses of large ensemble simulations and individual forcing experiments have strengthened the relationships between the seasonal advance of intense TCs and anthropogenic warming. We believe that the inclusion of these additional analyses has significantly improved the quality of the paper.

The authors also relate the trends in the seasonal occurrence of intense TCs to trends in extreme rainfall events. They note that, at least in South China, there is historically a local minima in extreme rainfall events in mid-summer, but that this time period has seen the largest increasing trend in extreme rainfall events. They attribute this to an increasing trend in extreme rainfall produced by intense TCs during that time period due to the shift towards having intense TCs earlier in the season.

Response

We appreciate the insight and thoughtfulness of the referee's comments. To address the concerns of the referee, we have significantly revised the statements regarding extreme rainfall events (please see below). We have also added the analysis for the Gulf of Mexico, another region heavily impacted by TCs. Consistent with South China, an increasing trend in extreme rainfall events is observed in the Gulf of Mexico in mid-summer, associated with an increasing trend in extreme rainfall produced by

intense TCs (see below). We have restricted our conclusion regarding extreme rainfall events to these two regions in the revised manuscript and will investigate its generalizability in the future.

The methods are generally sound, with appropriate use of statistics and quantification of uncertainty. The paper is well-written. However, while I think this paper is worthy of publication, there are some revisions that should be made before it can be published. Some of the conclusions seem a bit overstated and some methods need to be clarified.

Response

Thank you for your insightful comments and feedback on our manuscript. Following your suggestion, we have conducted numerous additional analyses to better support the detection and attribution conclusions and revised the statements regarding extreme rainfall events. Additional explanations have been included in Methods to enhance clarity. We believe that the suggested revisions have made the manuscript stronger and clearer. If there are remaining questions concerning the revised manuscript, we look forward to addressing them.

Comments

1. The most closely related study is that of Truchelut et al 2022 which found a trend towards earlier onset of the TC season in the North Atlantic in association with changes in environmental conditions. However in this study, when all TC events are considered (extended data Figure 1), no shifts towards earlier in the season are identified. There are a variety of different factors that could explain this - looking at the whole Northern Hemisphere versus just the North Atlantic, computing the trends in number of TC events on a monthly basis (I think)? versus finer temporal resolution - but the authors should explicitly relate their results to the Truchelut study and discuss the differences.

Response

The referee brings up an excellent point. A recent study (Truchelut et al., 2022) indicates a trend towards the earlier onset of the beginning time of TC season, in terms of the initial formation dates of the first TC and the 1st-3rd percentiles of accumulated cyclone energy threshold dates, in the North Atlantic basin based on the best track data, which has been linked to warming ocean temperatures in spring. Truchelut et al. (2022) also indicate that there is no significant seasonal change in total TC activity in the North Atlantic basin, which is consistent with our finding at the hemispheric scale that there is no significant change in the seasonal occurrence of all TC events. We have added a sentence in Lines 150-151 to discuss the relation and difference between our results and the findings by Truchelut et al. (2022).

To further confirm, here, the shifting rates of total TC activity in the North Atlantic basin are examined, in terms of the median value of occurrence date, based on the ADT-HURSAT dataset used in our study. There is no significant trend in the median value of the seasonal occurrence of all TC events in the North Atlantic basin. Neither for the less intense TCs. However, when we only consider the intense TCs, an abrupt change in the earlier-shifting rates in the North Atlantic basin is observed (Figure R1), consistent with the results in the Northern Hemisphere (as shown in Figure 2b in the manuscript). Since the number of less intense TC events is much larger relative to intense TCs in the North Atlantic basin, there is no significant change when all TCs are considered.

To clarify, we have revised the relevant statements in the Introduction: “A recent study (Truchelut et al., 2022) identified an earlier onset of the beginning time of TC season, in terms of the initial formation dates, in the North Atlantic basin associated with a warming ocean in spring, but there is no seasonal change in total Atlantic TC activity.”

Figure R1. Linear trends of the occurrence date of TC events with different intensities during the active season in the North Hemisphere (NH; black), and North Atlantic (NA; red) basin. The dots correspond to the significant shifting trends at the 95% confidence level based on the Mann-Kendall test, and the open circles correspond to the insignificant trends. The shadings represent the 95% confidence intervals in the linear regression analysis.

References

Truchelut, R. et al. Earlier onset of North Atlantic hurricane season with warming oceans. *Nat. Commun.* **13**, 4646 (2022).

2. Line 261, Lines 218-221: The conclusion that changes in oceanic conditions are mainly caused by external forcing is not as well-supported as it could be. This seems to be based on the fact that the trends towards earlier onset of high percentile PI and OHC are seen also in the CMIP6 historical simulations and in future projections. But this does not directly imply attribution! Are the authors arguing that the CMIP6 multi-model mean averages out internal variability and reveals the forced response, and so since we see similar trends to observations, that indicates the observed trends are due to the forced response? If so, that needs to be made clear (for example, is it the multi-model mean that is being plotted?). A stronger attribution statement could be made by using a large ensemble (so that internal variability could be averaged out) in combination with historical versus natural-only simulations.

Response

We thank the referee for this insightful comment. To address the referee's concerns, we have made great efforts to conduct additional analyses to better support the detection and attribution conclusions. Following the referee's suggestion, we have estimated the seasonal changes of the high percentiles of PI and OHC based on the Community Earth System Model Version 2 (CESM2; Rodgers et al., 2021) large ensemble. In Figure R2, the high percentiles of PI and OHC show significant trends towards earlier onset, in agreement with these based on the multi-model mean of the CMIP6 historical simulations. To make a stronger attribution statement, we have performed this analysis based on the Detection and Attribution Model Intercomparison Project (DAMIP) experiments under greenhouse gas (GHG) forcing only, natural forcing only, and anthropogenic aerosol forcing only, respectively. As a part of CMIP6, DAMIP was designed to estimate the contributions of anthropogenic and natural forcings (Gillett et al., 2016). Based on the experiments under GHG forcing (Figure 3i-l in the revised manuscript; also shown as Figure R3 a-d here), there is an increase in the high PI rate and high OHC rate in the early season and a decrease in the late season. In contrast, no significant change in the seasonal cycle of favorable oceanic conditions is found under natural forcing (Extended Data Figure 9a-d in the revised manuscript; also shown as Figure R3 e-h here). Under anthropogenic aerosol forcing (Extended Data Figure 9e-h

in the revised manuscript; also shown as Figure R3 i-l here), the high PI rate exhibits an increase in March (late season) in the SH, and the high OHC rate exhibits an increase in November (late season) in the NH and in March (late season) in the SH, which slightly offsets the earlier-shifting trends induced by GHG forcing. These results suggest that the earlier onset of the high PI rate and high OHC rate is primarily driven by GHG forcing, while the contributions of the natural forcing and anthropogenic aerosols are negligible or even negative.

As the manuscript is substantially revised, we will not list all the quoted modifications here but instead, we will highlight some major ones.

- a) Please see the abstract for a great summary of our detection and attribution statements, “Using simulations from multiple global climate models, large ensemble, and individual forcing experiments, the earlier onset of favorable oceanic conditions is detectable and is found to be primarily driven by greenhouse gas forcing.”
- b) We have added several sentences in the Conclusion as follows: “The earlier onset of favorable oceanic conditions is detectable based on both the multi-model mean of the CMIP6 historical simulations and the CESM2 large ensemble. Using the DAMIP experiments in which individual forcing is prescribed, it is found that the earlier onset of favorable oceanic conditions is primarily driven by GHG forcing, while the contributions of the natural and anthropogenic aerosol forcings are negligible or even negative.”
- c) The results, figures, and methods are updated accordingly.

Thanks for your suggestion, as the inclusion of the additional analyses of the CESM2 large ensemble simulations and DAMIP experiments has strengthened the relationships between the seasonal advance of intense TCs and anthropogenic warming.

Figure R2. Changes in the seasonal cycle of the oceanic conditions obtained from the CMIP6 historical simulations and the CESM2 large ensemble. For (up) the multi-model mean of the CMIP6 historical simulations and (bottom) the CESM2 large ensemble. **a, b**, The CMIP6 multi-model mean of the climatological value of PI (solid line) and the linear trend of high PI rate (bar; the fractional proportion of area with the PI value above its 95th percentile relative to the tropical ocean area) in each month during the active season from 1981 to 2017 in the (a) NH, and (b) SH. The error bars indicate the standard deviation. **c, d**, The CMIP6 multi-model mean of the climatological value of OHC (solid line) and the linear trend of high OHC rate (bar; the fractional proportion of area with the OHC value above its 95th percentile relative to the tropical ocean area) in each month during the active season from 1981 to 2017 in the (c) NH, and (d) SH. **e–h**, Same as in **a–d**, respectively, but obtained from the CESM2 large ensemble.

Figure R3. Changes in the seasonal cycle of oceanic conditions obtained from the DAMIP experiments under the greenhouse gas (GHG) forcing only, natural (NAT) forcing only, and anthropogenic aerosol (AER) forcing only. For (up) the multi-model mean of experiments forced by the GHG, (middle) experiments forced by the NAT, and (bottom) experiments forced by the AER. **a, b**, The climatological value of PI (solid line) and the linear trend of high PI rate (bar) in each month obtained from the GHG experiments during the active season from 1981 to 2017 in the (a) NH, and (b) SH. The error bars indicate the standard deviation. **c, d**, The climatological value of OHC (solid line) and the linear trend of high OHC rate (bar) in each month obtained from the GHG experiments during the active season from 1981 to 2017 in the (c) NH, and (d) SH. **e–h**, Same as in **a–d**, respectively, but obtained from the NAT experiments. **i–l**, Same as in **a–d**, respectively, but obtained from the AER experiments.

References

- Rodgers, K. et al. Ubiquity of human-induced changes in climate variability. *Earth Syst. Dynam.* **12**, 1393–1411 (2021).
- Gillett, N. et al. (2016). The detection and attribution model intercomparison project (DAMIP v1.0) contribution to CMIP6. *Geosci. Model Dev.* **9**, 3685–3697 (2016).

3. Lines 262-264, Lines 241-243, lines 61-65: The earlier seasonal occurrence of extreme rainfall produced by intense TCs is used to argue that the earlier seasonal occurrence of intense TCs increases the risk of compound disasters by causing more overlap between intense TCs and other extreme events. However, this conclusion seems like a bit of a reach. If the intense TCs are the event causing the changes in extreme rainfall, then that is not necessarily a compound disaster. Just because the shift in intense TCs might be “merging the double peaks in the seasonal cycle of extreme rainfall events” does not mean there is a compound event, per se. Changes in compound extremes are certainly highly impactful and important to understand, but doing so requires a careful definition of the compound event and an examination of changes in its occurrence specifically, rather than just a general examination of changes in the seasonal distribution of extreme rainfall and of intense TCs. Instead, the authors could state that the earlier onset of intense TCs also leads to an earlier onset in extreme rainfall associated with them, and this could contribute to extreme rainfall occurring during unexpected times of year, or something like that.

Response

We agree with the referee’s comment on the seasonal occurrence of extreme rainfall. Following the constructive suggestion, we have revised the text in the Conclusion as follows: “In the South China and Gulf of Mexico, the earlier onset of intense TCs contributes substantially to an earlier onset in extreme rainfall, which could contribute to the increased overlap period between the extreme rainfall produced by the summer monsoon system and that produced by intense TCs, thus leading to devastating impacts that are well beyond any one of these events individually (Zscheischler et al.,

2018; Patricola et al., 2022; Zhu et al., 2022).” Note that the Gulf of Mexico has been included as another example in the revised manuscript and similar conclusions can be drawn as for South China (please see below).

To further reveal the seasonal cycle changes in extreme rainfall events, a simple definition of the persistent rainfall event by Chen & Zhai (2013) is applied. An increasing trend of the annual number of persistent rainfall events in South China during July-September is observed (included as Supplementary Figure 3a in the revised manuscript; also shown in Figure R4b here), which is consistent with the more overlap between extreme rainfall produced by the summer monsoon system and that produced by intense TCs. This point has been reflected as supporting evidence. A few sentences have been added to address the limitations.

Figure R4. The earlier seasonal occurrence of extreme rainfall produced by intense TCs in South China. **a**, The seasonal distribution of the total number of all extreme rainfall events (black line) and the extreme rainfall events induced by intense TCs (blue line) during 1981–2017 in South China (105-120°E; 15-25°N), and the linear trends (bars) of the annual number in each month. The diagonal lines indicate significance at the 95% confidence level based on the Mann-Kendall test. The error bars indicate the 95% confidence intervals in the linear regression analysis. **b**, Time series (dashed line) of the annual number of persistent rainfall events and its linear trend (solid line) in South China during July- September from 1981 to 2017. The shading represents the 95% confidence interval in the linear regression analysis.

References

- Chen, Y. & Zhai, P. Persistent extreme precipitation events in China during 1951–2010. *Clim. Res.* **57**, 143–155 (2013).
- Patricola, C., Cassidy, D. & Klotzbach, K. Tropical oceanic influences on observed global tropical cyclone frequency. *Geophys. Res. Lett.* **49**, e2022GL099354 (2022).
- Zhu, Y., Collins, J., Klotzbach, P. & Schreck III, C. Hurricane Ida (2021): rapid intensification followed by slow inland decay. *Bull. Am. Meteorological Soc.* **103**, E2354–E2369 (2022).
- Zscheischler, J. et al. Future climate risk from compound events. *Nat. Clim. Change* **8**, 469–477 (2018).

4. Lines 239-241: An example for South China is given. Is it generalizable to other regions? Be careful about making broad conclusions based on one specific region.

Response

We thank the referee for highlighting the issue. In the revised manuscript, we have added the analysis for the Gulf of Mexico, another region heavily impacted by TCs (Klotzbach et al., 2018; Marsooli et al., 2019; Zhu et al., 2022). Consistent with South China, an increasing trend in extreme rainfall events is observed during July-September in the Gulf of Mexico, associated with an increasing trend in extreme rainfall produced by intense TCs (included as Figure 4b in the revised manuscript; also shown in Figure R5a here). Furthermore, there is also an increasing trend in the annual number of persistent rainfall events in the Gulf of Mexico during that time period (included as Supplementary Figure 3b in the revised manuscript; also shown as Figure R5b here).

In the revised manuscript, we have restricted our conclusions regarding extreme rainfall events to the two regions, i.e., South China and the Gulf of Mexico, and will investigate its generalizability in a future study.

Figure R5. The earlier seasonal occurrence of extreme rainfall produced by intense TCs in the Gulf of Mexico. a, The seasonal distribution of the total number of all extreme rainfall events (black line) and the extreme rainfall events induced by intense TCs (blue line) during 1981–2017 in the Gulf of Mexico (80-100°W; 25-35°N), and the linear trends (bars) of the annual number in each month. The diagonal lines indicate significance at the 95% confidence level based on the Mann-Kendall test. The error bars indicate the 95% confidence intervals in the linear regression analysis. **b,** Time series (dashed line) of the annual number of the persistent rainfall events and its trend (solid line) in Gulf of Mexico during July-September from 1981 to 2017. The shading represents the 95% confidence interval in the linear regression analysis.

References

- Klotzbach, P., Bowen, S., Pielke Jr. R. & Bell, M. Continental U.S. Hurricane Landfall Frequency and Associated Damage: Observations and Future Risks. *Bull. Am. Meteorological Soc.* **99(7)**, 1359–1376 (2018).
- Marsooli, R., Lin, N., Emanuel, K. & Feng, K. Climate change exacerbates hurricane flood hazards along US Atlantic and Gulf Coasts in spatially varying patterns. *Nat. Commun.* **10**, 3785 (2019).
- Zhu, Y., Collins, J., Klotzbach, P. & Schreck III, C. Hurricane Ida (2021): rapid intensification followed by slow inland decay. *Bull. Am. Meteorological Soc.* **103**, E2354–E2369 (2022).

5. Lines 194-196, Lines 331-335: It is not clear exactly how the trends in the occurrence of the favorable environmental conditions are being computed. Is the insinuation that if a larger spatial area has PI greater than the 95th percentile relative to the whole tropics, that more RI events will occur? It is not clear. Also, why should the 95th percentile of PI or OHC necessarily correspond to the 95th percentile of TC intensity changes? This choice should be justified by a more careful analysis of the environmental conditions that lead to RI, as performed by (for example) Bhatia et al 2022.

Response

We thank the referee for guiding us to deepen the analysis of the favorable environmental conditions for RI events. We agree that RI events (exceeding 35 knots over a period of 24 h in our study; approximately 95th percentile of TC intensity changes) should not strictly correspond to the 95th favorable environmental conditions, as it is well known that RI is affected by both the environmental conditions and internal dynamical processes (Hendricks et al., 2010; Kowch & Emanuel, 2015).

To address this issue, we have calculated the environmental thresholds for RI events following the method by Bhatia et al. (2022). Consistent with Bhatia et al. (2022), we found that stronger PI, higher OHC, higher RH, and weaker VWS are all beneficial for RI (Table REDACTED). The exact value differs from that by Bhatia et al. (2022), as we use 35kt as the threshold of RI definition rather than the 30kt used by Bhatia et al. (2022). This result suggests that it is reasonable to use the high quartiles of PI, OHC, and RH and the low quartiles of VWS to indicate the environmental conditions that lead to RI. In the main text, the 95th of PI, OHC, and RH, and the 5th of VWS are used. A sensitivity analysis has been conducted, in which the 90th of PI, OHC, and 10th of VWS are used. Consistently, an earlier shift of the seasonal cycle favorable oceanic conditions is obvious (included as Supplementary Figure 4 in the revised manuscript; also shown as Figure R6 here), while no significant change in the atmospheric conditions (included as Supplementary Figure 5 in the revised manuscript; also shown as Figure R7 here). Our original conclusions still hold and we believe these additional analyses have further strengthened our results. We hope that these changes will satisfactorily address your concerns.

Figure R6. Seasonal trends of the 90th percentile of oceanic factors. a, b, The climatological mean value of potential intensity (PI; solid line) and the linear trend of high PI rate (bar; the fractional proportion of area with the PI value above its 90th percentile relative to the tropical ocean area) in each month during the active season from 1981 to 2017 obtained from the ECMWF monthly Ocean Reanalysis database in the (a) NH, and (b) SH. The error bars represent 95% confidence intervals in the linear regression analysis. **c, d,** The climatological mean value of ocean heat content (OHC; solid line) and the linear trend of high OHC rate (bar; the fractional proportion of area with the OHC value above its 90th percentile relative to the tropical ocean area) in each month during the active season from 1981 to 2017 the ECMWF monthly Ocean Reanalysis database in the (c) NH, and (d) SH.

Figure R7. Seasonal trends of the 90th percentile of relative humidity (RH) and the 10th percentile of vertical wind shear (VWS). **a, b,** The climatological mean value of RH (solid line) and the linear trend of high RH rate (bar; the fractional proportion of area with the RH value above its 90th percentile relative to the tropical ocean area) in each month during the active season from 1981 to 2017 obtained from the ERA-5 reanalysis dataset in the **(a)** NH, and **(b)** SH. The error bars represent 95% confidence intervals in the linear regression analysis. **c, d,** The climatological mean value of VWS (solid line) and the linear trend of low VWS rate (bar; the fractional proportion of area with the VWS value below its 10th percentile relative to the tropical ocean area) in each month during the active season from 1981 to 2017 obtained from the ERA-5 reanalysis dataset in the **(c)** NH, and **(d)** SH.

[REDACTED]

References

- Bhatia, K. et al. A potential explanation for the global increase in tropical cyclone rapid intensification. *Nat. Commun.* **13**, 6626 (2022).
- Hendricks, E. A., Peng, M. S., Fu, B. & Li, T. Quantifying environmental control on tropical cyclone intensity change. *Monthly Weather Rev.* **138**, 3243–3271 (2010).
- Kowch, R., & Emanuel, K. Are special processes at work in the rapid intensification of tropical cyclones? *Monthly Weather Rev.* **143**(3), 878–882 (2015).

6. Lines 38-39: While the cited study (Chand et al 2022) does suggest a declining number of TCs over the past century (by explicitly tracking TCs in 20th century reanalysis), other methodologies show different results (Emanuel, K., 2021: Atlantic tropical cyclones downscaled from climate reanalyses show increasing activity over past 150 years. Nature Communications, 12, 7027, <https://doi.org/10.1038/s41467-021-27364-8>), which should be acknowledged.

Response

Thank you for the valuable recommendation. We have added this important point and relevant references to the Introduction “This warming may have already impacted tropical cyclone activity at a global scale (Emanuel 2020; Kossin et al 2020; Chand et al 2022). Although changes in the annual number of global TCs are controversial (Emanuel 2020; Emanuel 2021; Chand et al 2022), an increasing trend in the number of global intense TCs has been noted (Kossin et al 2020; Emanuel 2021), and significant efforts devoted to reducing uncertainties in data quality issues (Vecchi & Knuston 2008; Emanuel 2013).”

References

- Chand, S. et al. Declining tropical cyclone frequency under global warming. *Nat. Clim. Change* **12**, 655–661 (2022).
- Emanuel, K. Downscaling CMIP5 climate models shows increased tropical cyclone activity over the 21st century. *Proc. Natl Acad. Sci. USA* **110**, 12219–12224 (2013).

Emanuel, K. Response of global tropical cyclone activity to increasing CO₂: results from downscaling CMIP6 models. *J. Clim.* **34**, 57–69 (2020).

Emanuel, K. Atlantic tropical cyclones downscaled from climate reanalyses show increasing activity over past 150 years. *Nat. Commun.* **12**, 7027 (2021).

Kossin, J., Knapp, K., Olander, T. & Velden, C. Global increase in major tropical cyclone exceedance probability over the past four decades. *Proc. Natl Acad. Sci. USA* **117**, 11975–11980 (2020).

Vecchi, G. & Knutson, T. On estimates of historical North Atlantic tropical cyclone activity. *J. Clim.* **21**, 3580–3600 (2008).

7. Figure 1 and other similar figures: What exactly is plotted on the line and the y-axis? Is it the total number of events in that month across the entire record? The numbers are too high to represent the average number of events in that month per year. Please clarify.

Response

Yes, you are right. It is the total number of events. We have revised the label and caption of Figure 1 to clarify this.

8. Figure 1c: How are these trends calculated spatially? Is the intraseasonal difference in intense TC number computed for each grid point? And if so, what grid is used?

Response

Yes, the intraseasonal difference in intense TC number is computed for each grid point. We originally used the grid with a 10° × 10° resolution, and we have now changed to a finer grid with a 5° × 5° resolution, following the suggestion of referee #2. The figure has been updated.

9. Figure 2 b,e: I know statistical significance is indicate by the symbol type, but could error bars be added to these plots?

Response

Thanks! Following the suggestion, we have added error bars to show the 95% confidence intervals for the linear trend estimates in Figure 2 b, e.

10. Figure 2 c,f: Is it also the median value of RI occurrence date that is plotted?

Response

Yes, it is the median value of the RI occurrence date. We have revised the caption to clarify this.

11. Line 117: Suggest specifying open circles correspond to the insignificant trends

Response

We have changed it to “open circles” according to the suggestion.

12. Line 319-323: Why are these variables defined? They are not shown in any equation anywhere.

Response

Sorry for this confusion. We did describe the relevant variables for OHC but forgot to include its equation. We have now added it to the Methods.

13. Regarding the model simulations, what is being plotted in Figure 3? The multi-model mean?

Response

Thank you for pointing this out. Yes, it is the multi-model mean. We have revised the figure and caption to clarify this. Explanations have been added in the Methods.

14. Regarding the extreme rainfall events (above 50mm, which is stated to be the 95th percentile of daily rainfall) plotted for South China in Figure 4, the methods should indicate the specific region that is being considered.

Response

We originally estimated the seasonal trends of the occurrence of extreme rainfall in South China. We have added the analysis for the Gulf of Mexico in the revised manuscript. Following your suggestion, the definitions of South China (105-120°E; 15-25°N) and the Gulf of Mexico (80-100°W; 25-35°N) have been added to the Methods.

15. Extended data Figure 1: Should the y-axis label be TC number, rather than Intense TC number? The caption indicates that all TC events are being considered.

Response

Yes, you are right. The label has been revised accordingly. We have carefully read the revised manuscript to ensure that they are typo-free.

16. Extended data Figure 1 and Extended data Figure 2: Please provide a full description of the plot in the caption, rather than saying it is the same as Figure 1a,b.

Response

Thanks for the suggestion. We have added a full description of the plot in the

captions.

17. Extended data Figure 3: The caption refers to RI events in panels a,b as being plotted with a red line, but the plot shows a pink/purple line.

Response

Thanks. Revision has been made accordingly.

18. Line 329-330, In extended data Figure 6, the seasonal trends in the occurrence of high relative humidity and low vertical wind shear are compared in two different reanalysis datasets – ERA-5 and the NCEP/NCAR reanalysis. It is good that multiple reanalyses are considered, but the NCEP/NCAR reanalysis is somewhat outdated (and has known biases some fields that influence TCs, such as upper tropospheric/lower stratospheric temperatures), so it would be beneficial to compare to a newer reanalysis such as JRA-55, MERRA-2, or CFSR. In that case, the trends in potential intensity could also be compared in a second reanalysis.

Response

We appreciate that the referee thinks carefully and constructively about the use of the reanalysis datasets. We agree that the NCEP/NCAR reanalysis has large biases in some variables that influence TCs. In the revised version, we estimated the seasonal trends in the occurrence of high percentiles of relative humidity and low percentiles of vertical wind shear by using the ERA-5, JRA-55, and MERRA-2 reanalysis datasets. Consistent with the results based on the ERA-5 reanalysis dataset (Extended Data Figure 5 in the manuscript), there is no significant change in the seasonal occurrence of high percentiles of relative humidity and low percentiles of vertical wind shear based on the JRA-55 and MERRA-2 reanalysis datasets, as shown in Figure R8 and Figure R9, respectively. These are reflected in Extended Data Figures 6 & 7 in the revised manuscript. A paragraph has been added in the Methods to describe these reanalysis datasets.

Thank you for this suggestion, as the inclusion of more reliable reanalysis datasets

has increased the confidence in our analysis.

Figure R8. Observed changes in the seasonal cycle of atmospheric conditions obtained from the JRA-55 reanalysis dataset. a, b, The climatological mean value of relative humidity (RH; solid line) and the linear trend of high RH rate (bar) in each month during the active season from 1981 to 2017 obtained from the JRA-55 reanalysis dataset in the (a) NH, and (b) SH. The error bars indicate the 95% confidence intervals in the linear regression analysis. **c, d,** The climatological mean value of vertical wind shear (VWS; solid line) and the linear trend of low VWS rate (bar) in each month during the active season from 1981 to 2017 obtained from the JRA-55 reanalysis dataset in the (c) NH, and (d) SH.

Figure R9. Observed changes in the seasonal cycle of atmospheric conditions obtained from the MERRA-2 reanalysis dataset. a, b, The climatological mean value of relative humidity (RH; solid line) and the linear trend of high RH rate (bar) in each month during the active season from 1981 to 2017 obtained from the MERRA-2 reanalysis dataset in the (a) NH, and (b) SH. The error bars indicate the 95% confidence intervals in the linear regression analysis. **c, d,** The climatological mean value of vertical wind shear (VWS; solid line) and the linear trend of low VWS rate (bar) in each month during the active season from 1981 to 2017 obtained from the MERRA-2 reanalysis dataset in the (c) NH, and (d) SH.

19. Extended data Figure 7: For clarity, the caption should indicate which future scenario the CMIP6 simulations are using to arrive at the 2080-2099 period.

Response

Thanks! The future high-emission scenario SSP585 is used. We have revised the caption to clarify this.

Author Response to Referee#2 of “Seasonal Advance of Intense Tropical Cyclones in A Warming Climate”

Recommendation

This is an interesting study of seasonality change in intense tropical cyclones, with potentially important implications for understanding compound hazards in populous regions, a key component of near-future climate risk. In my view, the manuscript is generally suited to the readership of *Nature*, focussing on a sufficiently important topic, but is not publishable in its current form. The authors have identified consistent seasonal changes in area-mean oceanic environmental quantities and tropical cyclones, which is an important and novel observation, although other studies have examined similar environmental metrics in relation to TCs. However, the current manuscript falls short of definitively linking oceanic changes with tropical cyclone activity. The introduction refers to detection and attribution of shifts in TC behaviour to climate change, but the paper does not present any such analysis. Additionally, the North Atlantic, where no seasonal changes are identified in the authors’ analysis, is largely not discussed. These issues are explained below, and the authors should consider addressing these and resubmitting to *Nature* or, perhaps more appropriately, *Nature Climate Change*.

Response

Thanks for your insightful comments and encouragement on our manuscript. We have carefully considered and addressed these main comments in the revised version of the manuscript.

- a) We totally agree that both atmospheric and oceanic factors have an impact on the seasonal activity of intense TCs / RI events (Klotzbach, 2012; Zhao et al., 2018; Klotzbach et al., 2019; Jones et al., 2020; Patricola et al., 2022). To address the concerns of the referee, we have added an additional analysis of the effect of the atmospheric factors. In the revised manuscript, we estimated the seasonal trends in the occurrence of the high percentiles of relative humidity and low percentiles of vertical wind shear by using three atmospheric

reanalysis datasets (ERA-5, JRA55, and MERRA2). Consistently, there is no significant change in the seasonal occurrence of atmospheric factors in all three datasets. In contrast, the high percentiles of oceanic factors (potential intensity and ocean heat content) show significant trends towards earlier onset based on two oceanic reanalysis datasets. In this case, the earlier onset of favorable oceanic conditions contributes most to the earlier onset of RI events and thus the earlier onset of intense TCs;

- b) We have made great efforts to conduct additional analyses to better support the detection and attribution conclusions. We have estimated the seasonal changes of the high percentiles of oceanic factors based on the Community Earth System Model Version 2 (CESM2) large ensemble, which contains 100 members. The internal variability could be largely suppressed when we take the ensemble mean. It is further confirmed that favorable oceanic conditions show significant trends towards earlier onset based on the CESM2 large ensemble. To make a stronger attribution statement, we have repeated the analysis based on the Detection and Attribution Model Intercomparison Project (DAMIP) experiments under the greenhouse gas forcing only, natural forcing only, and anthropogenic aerosol forcing only, respectively. Results suggest that the earlier onset of favorable oceanic conditions is primarily driven by the greenhouse gas forcing, while the contributions of the natural and anthropogenic aerosol forcings are negligible or even negative. Overall, the inclusion of the additional analyses of large ensemble simulations and individual forcing experiments has strengthened the relationships between the seasonal advance of intense TCs and anthropogenic warming;
- c) We have provided a detailed analysis of the seasonal changes in intense TCs over the North Atlantic basin. A significant trend of the median value of the occurrence of intense TCs is observed over the entire North Atlantic basin. Based on the analysis of the climatological distribution of intense TCs and its inter-seasonal difference between the early and late seasons over the North Atlantic basin, it is found that intense TCs (as well as RI events) are mostly concentrated in the Gulf of Mexico and the western part of the North Atlantic basin, consistent with the spatial pattern of the inter-seasonal difference of intense TCs. Furthermore, as identified by Klotzbach et al. (2022), there is a

large increase in PI in the Gulf of Mexico and the western part of the North Atlantic basin, while changes in the eastern part are much weaker or even negative. It is suggested that the spatial pattern of the inter-seasonal difference of intense TCs over the North Atlantic basin is also consistent with the changes in thermodynamic factors. We have added a few sentences to discuss the seasonal changes of intense TC occurrence over the North Atlantic basin in the main text.

We believe that the major revisions undertaken have significantly improved the quality of the paper. Detailed and specific responses to the referees' comments are provided below, following each comment. If there are remaining questions concerning the revised manuscript, we look forward to addressing them.

References

- Jones, J., Bell, M. & Klotzbach, P. Tropical and subtropical North Atlantic vertical wind shear and seasonal tropical cyclone activity. *J. Clim.* **33**, 5413–5426 (2020).
- Klotzbach, P. El Niño-Southern Oscillation, the Madden-Julian Oscillation and Atlantic basin tropical cyclone rapid intensification. *J. Geophys. Res.* **117**, D1410 (2012).
- Klotzbach, P. et al. Trends in global tropical cyclone activity: 1990–2021. *Geophys. Res. Lett.* **49**, e2021GL095774 (2022).
- Klotzbach, P. et al. Seasonal tropical cyclone forecasting. *Trop. Cyclone Res. Rev.* **10**, 134–149 (2019).
- Patricola, C., Cassidy, D. & Klotzbach, P. Tropical oceanic influences on observed global tropical cyclone frequency. *Geophys. Res. Lett.* **49**, e2022GL099354 (2022).
- Zhao, H., Duan, X., Raga, G. & Klotzbach, P. Changes in characteristics of rapidly intensifying western North Pacific tropical cyclones related to climate regime shifts. *J. Clim.* **31**, 8163–8179 (2018).

Major comments

1. Oceanic vs atmospheric factors in explaining earlier-shifting trends.

The authors logically look to explain the earlier-shifting trends in intense TCs / RI events by analysing oceanic environmental factors (PI and OHC) and atmospheric factors (wind shear and humidity). The authors justify focussing on oceanic factors by appealing to their shared seasonality with intense TCs, and this analysis is strengthened by the inclusion of two oceanic datasets. However, the atmospheric factors are not as well explained. Firstly, the reanalysis and models are not in agreement and, secondly, some significant trends are contrary to the oceanic results. Changes in wind shear across basins can be induced by global teleconnections, which in turn modulate seasonal TC activity, so the authors should not ignore the atmospheric factors.

Response

We appreciate the insight and thoughtfulness of the referee's comments and suggestions. We totally agree that both atmospheric and oceanic factors have an impact on the seasonal activity of intense TCs / RI events (Klotzbach, 2012; Zhao et al., 2018; Klotzbach et al., 2019; Jones et al., 2020; Patricola et al., 2022).

To address the concerns of the referee, we have added more analyses of the atmospheric factors. In the revised manuscript, we estimated the seasonal trends in the occurrence of the high percentiles of relative humidity and low percentiles of vertical wind shear by using the ERA-5, JRA-55, and MERRA-2 reanalysis datasets. Consistent with the results based on the ERA-5 reanalysis dataset (Extended Data Figure 5 in the manuscript; also Figure R1 here), there is no significant change in the seasonal occurrence of the high percentiles of relative humidity and low percentiles of vertical wind shear based on the JRA-55 and MERRA-2 reanalysis datasets, as shown in Figure R2 & R3, respectively. These results are reflected in Extended Data Figures 6 & 7 in the revised manuscript. In contrast, the high percentiles of PI and OHC show significant trends towards earlier onset based on two oceanic datasets. Hence, the earlier onset of favorable oceanic conditions contributes most to the earlier onset of the occurrence of RI events.

In the revised manuscript, We have added a few sentences to discuss the roles of

oceanic and atmospheric factors in explaining earlier-shifting trends “While the atmospheric factors (i.e., relative humidity and vertical wind shear) were previously documented to impact RI events and widely applied as forecasting variables (Klotzbach, 2012; Klotzbach et al., 2019; Jones et al., 2020), the seasonal cycle of the atmospheric factors does not show a significant change based on three atmospheric reanalysis datasets (Extended Data Figures 5, 6 and 7). Hence, the observed earlier onset of favorable oceanic conditions contributes most to the earlier onset of RI events and thus that of intense TCs, while changes in the atmospheric conditions contribute little.”

For clarity, we have also added one sentence in the Conclusion as follows: “In contrast, the changes in the atmospheric factors (i.e., relative humidity and vertical wind shear) are either modest or insignificant, and thus do not contribute much to this earlier-shifting trend.” The abstract, results, figures, and methods are updated accordingly. We have also removed the statements regarding the comparison of the climatological seasonality of intense TCs and the related environmental factors.

Regarding the comment about the disagreement between the reanalysis and models, we did not mean to compare the variations in atmospheric factors obtained from the reanalysis and models. In fact, the left panel shows the results obtained from the ERA-5 reanalysis dataset, and the right panel shows the results obtained from the NCEP/NCAR reanalysis dataset (Extended Data Figure 6 in the first version of the manuscript). The inconsistency between the two panels is from the differences in the ERA-5 and NCEP/NCAR reanalysis datasets. As suggested by referee #1, the NCEP/NCAR reanalysis has large biases in some variables that influence TCs. Following this suggestion, we have used the JRA-55 and MERRA-2 reanalysis datasets to replace the NCEP/NCAR reanalysis dataset in the revised manuscript. Despite some differences in the trends for individual months in these three datasets, all three datasets consistently show no significant seasonal variation in either relative humidity or vertical wind shear, which strengthens our confidence in the results. For clarity, the results obtained from the ERA-5, JRA-55, and MERRA-2 reanalysis datasets are shown in Extended Data Figures 5, 6, and 7, respectively, in the revised manuscript.

Our original conclusions still hold and we believe the analysis of atmospheric factors based on three datasets and literature review (Klotzbach, 2012; Zhao et al., 2018; Klotzbach et al., 2019; Jones et al., 2020; Patricola et al., 2022) have significantly improved the quality of the paper. We hope that these changes will satisfactorily address

your concerns.

Figure R1. Observed changes in the seasonal cycle of atmospheric conditions obtained from the ERA-5 reanalysis dataset. a, b, The climatological mean value of relative humidity (RH; solid line) and the linear trend of high RH rate (bar) in each month during the active season from 1981 to 2017 obtained from the ERA-5 reanalysis dataset in the (a) NH, and (b) SH. The error bars indicate 95% confidence intervals in the linear regression analysis. **c, d,** The climatological mean value of vertical wind shear (VWS; solid line) and the linear trend of low VWS rate (bar) in each month during the active season from 1981 to 2017 obtained from the ERA-5 reanalysis dataset in the (c) NH, and (d) SH.

Figure R2. Observed changes in the seasonal cycle of atmospheric conditions obtained from the JRA-55 reanalysis dataset. a, b, The climatological mean value of relative humidity (RH; solid line) and the linear trend of high RH rate (bar) in each month during the active season from 1981 to 2017 obtained from the JRA-55 reanalysis dataset in the (a) NH, and (b) SH. The error bars indicate 95% confidence intervals in the linear regression analysis. **c, d,** The climatological mean value of vertical wind shear (VWS; solid line) and the linear trend of low VWS rate (bar) in each month during the active season from 1981 to 2017 obtained from the JRA-55 reanalysis dataset in the (c) NH, and (d) SH.

Figure R3. Observed changes in the seasonal cycle of atmospheric conditions obtained from the MERRA-2 reanalysis dataset. a, b, The climatological mean value of relative humidity (RH; solid line) and the linear trend of high RH rate (bar) in each month during the active season from 1981 to 2017 obtained from the MERRA-2 reanalysis dataset in the (a) NH, and (b) SH. The error bars indicate 95% confidence intervals in the linear regression analysis. **c, d,** The climatological mean value of vertical wind shear (VWS; solid line) and the linear trend of low VWS rate (bar) in each month during the active season from 1981 to 2017 obtained from the MERRA-2 reanalysis dataset in the (c) NH, and (d) SH.

References

Jones, J., Bell, M. & Klotzbach, P. Tropical and subtropical North Atlantic vertical wind shear and seasonal tropical cyclone activity. *J. Clim.* **33**, 5413–5426 (2020).

Klotzbach, P. El Niño-Southern Oscillation, the Madden-Julian Oscillation and Atlantic basin tropical cyclone rapid intensification. *J. Geophys. Res.* **117**, D1410 (2012).

Klotzbach, P. et al. Seasonal tropical cyclone forecasting. *Trop. Cyclone Res. Rev.* **10**, 134–149 (2019).

Patricola, C., Cassidy, D. & Klotzbach, P. Tropical oceanic influences on observed global tropical cyclone frequency. *Geophys. Res. Lett.* **49**, e2022GL099354 (2022).

Zhao, H., Duan, X., Raga, G. & Klotzbach, P. Changes in characteristics of rapidly intensifying western North Pacific tropical cyclones related to climate regime shifts. *J. Clim.* **31**, 8163–8179 (2018).

2. Model shortcomings.

The authors should be more critical of the relatively low-resolution CMIP6 models and their current shortcomings in analysis of TC environments in models. The ocean mesoscale, which can be important for TC intensification, is not represented in CMIP6 models. While I understand that the authors focus on large-scale environmental fields, which are represented reasonably in some models, there are improvements in sea surface temperature biases and large-scale atmospheric circulation in higher-resolution models, such as HighResMIP, which the authors should consider exploiting to strengthen their analysis.

Response

Thank you very much for pointing us to this important issue. We admit that there are some biases in simulating ocean mesoscale conditions of the relatively low-resolution CMIP6 models, which may introduce uncertainty into the analysis of TC-related environmental field (Hsu et al., 2019; Huang et al., 2021; Patricola et al., 2022). Though the large-scale environmental fields are the focus of our study, it would be better to make a comparison of the results obtained from the high-resolution and low-resolution model outputs of CMIP6 HighResMIP simulations. The HighResMIP multi-model ensemble analysis shows that the earlier onset of the high percentiles of PI and OHC is also evident in both the high-resolution and low-resolution model outputs (Extended Data Figure 10; also shown as Figure R4 here). We have also added a few sentences and references (Hsu et al., 2019; Huang et al., 2021; Patricola et al., 2022) to address this issue. Thank you for this constructive suggestion, as the consistency in the results based on the high-resolution and low-resolution model outputs of CMIP6

HighResMIP simulations has increased the confidence in our results based on the historical simulations from the relatively low-resolution CMIP6 models.

Figure R4. Seasonal trends in the oceanic conditions obtained from the CMIP6 HighResMIP simulations. a, b, The linear trend of high PI rate (the fractional proportion of area with the PI value above its 95th percentile relative to the tropical ocean area) in each month during the active season from 1981 to 2017 obtained from the high resolution (HighRes; grey bars) outputs and the low resolution (LowRes; blue bars) outputs in the (a) NH, and (b) SH. **c, d,** The linear trend of high OHC rate (the fractional proportion of area with the OHC value above its 95th percentile relative to the tropical ocean area) in each month during the active season from 1981 to 2017 obtained from the high resolution (grey bars) outputs and the low resolution (blue bars) outputs in the (c) NH, and (d) SH.

References

Hsu, W., Patricola, C. & Chang, P. The impact of climate model sea surface temperature biases on tropical cyclone simulations. *Clim. Dyn.* **53**, 173– 192 (2019).
Huang, H., Patricola, C. & Collins, W. The influence of ocean coupling on simulated

and projected tropical cyclone precipitation in the HighResMIP-PRIMAVERA simulations. *Geophys. Res. Lett.* **48**, e2021GL094801 (2021).

Patricola, C., Cassidy, D. & Klotzbach, P. Tropical oceanic influences on observed global tropical cyclone frequency. *Geophys. Res. Lett.* **49**, e2022GL099354 (2022).

3. Clarifications and improvements to the text.

In several places, phrasing and language are vague and the text requires revision using appropriate and concrete language (frequency, particular hazards etc). See line-by-line comments for further detail. Additionally, author names in references are not formatted consistently.

Response

We thank the referee for this useful comment. We have undertaken significant revisions of the text. We hope that these changes will satisfactorily address your concerns.

Line-by-line comments

Line 21: “globally”—not sure this is entirely true. The North Atlantic is an important exception.

Response

We thank the referee for pointing this out. Indeed, the significant shifting trend of intense TCs towards earlier onset is observed in the two hemispheres, though the regional differences are still evident. We modified the text to address this suggestion, “Here we identify a significant seasonal advance of intense TCs since the 1980s in most tropical oceans, with an earlier-shifting rate of 3.7 and 3.2 days per decade for the Northern and Southern Hemispheres, respectively.” We have used ‘in the two

hemispheres’ to replace ‘globally’ throughout the revised manuscript where applicable. We have also added a few sentences to discuss the regional differences of the earlier-shifting trend of intense TCs in the Results and Conclusion.

Note that there is a significant trend of the median value of the occurrence of intense TCs over the entire North Atlantic basin, with a rate of 0.8 days per decade (also shown in Extended Data Table 1). Furthermore, consistent with the results in the North Hemisphere, an abrupt change in the earlier-shifting rates is observed for intense TCs over the North Atlantic basin, but not for less intense TCs (Figure R5). This result provides further evidence that the North Atlantic basin is not an exception. A detailed analysis of the seasonal changes in intense TCs over the North Atlantic basin is presented below.

Figure R5. Linear trends of the occurrence date of TC events with different intensities during the active season in the North Hemisphere (NH; black), and North Atlantic (NA; red) basin. The dots correspond to the significant shifting trends at the 95% confidence level based on the Mann-Kendall test, and the open circles correspond to the insignificant trends. Shadings represent 95% confidence intervals in the linear regression analysis.

Lines 25–26: There could be multiple factors for the RI trend, so revise this statement.

Response

Thank you for this comment. We totally agree that both atmospheric and oceanic

factors have an impact on the seasonal activity of intense TCs / RI events (Klotzbach, 2012; Zhao et al., 2018; Klotzbach et al., 2019; Jones et al., 2020; Patricola et al., 2022).

In the revised manuscript, We have added a few sentences to discuss the roles of oceanic and atmospheric factors in explaining earlier-shifting trends “While the atmospheric factors (i.e., relative humidity and vertical wind shear) have previously been documented to impact RI events and widely applied as forecasting variables (Klotzbach, 2012; Klotzbach et al., 2019; Jones et al., 2020), the seasonal cycle of the atmospheric factors does not show a significant change based on three atmospheric reanalysis datasets (Extended Data Figures 5, 6 and 7). Hence, the observed earlier onset of the favorable oceanic conditions contributes most to the earlier onset of RI events and thus that of intense TCs, while changes in the atmospheric conditions contribute little.”

References

- Jones, J., Bell, M. & Klotzbach, P. Tropical and subtropical North Atlantic vertical wind shear and seasonal tropical cyclone activity. *J. Clim.* **33**, 5413–5426 (2020).
- Klotzbach, P. El Niño–Southern Oscillation, the Madden-Julian Oscillation and Atlantic basin tropical cyclone rapid intensification. *J. Geophys. Res.* **117**, D1410 (2012).
- Klotzbach, P. et al. Seasonal tropical cyclone forecasting. *Trop. Cyclone Res. Rev.* **10**, 134–149 (2019).
- Patricola, C., Cassidy, D. & Klotzbach, P. Tropical oceanic influences on observed global tropical cyclone frequency. *Geophys. Res. Lett.* **49**, e2022GL099354 (2022).
- Zhao, H., Duan, X., Raga, G. & Klotzbach, P. Changes in characteristics of rapidly intensifying western North Pacific tropical cyclones related to climate regime shifts. *J. Clim.* **31**, 8163–8179 (2018).

Lines 40–42: Other evidence suggests some poleward shift may be due to changes in seasonality (Feng et al., 2021)—this should be briefly included in the text.

Response

Thank you for this recommendation, we have added the reference Feng et al. (2021) to the Introduction “Poleward migration of the locations of TC activity was observed in most ocean basins in recent decades and is particularly robust in the western North Pacific basin, which is related to changes in TC seasonality (Feng et al., 2021).”

References

Feng, X., Klingaman, N. & Hodges, K. Poleward migration of western North Pacific tropical cyclones related to changes in cyclone seasonality. *Nat. Commun.* **12**, 6210 (2021).

Line 48: “serious consequences” is a bit vague. Be more specific and indicate particular hazards.

Response

Thanks. Following the referee’s comments, we have added this text to make sure it is clear to our readers: “Most regions lack resilience to intense TCs, which could cause many deaths and damage to property through destructive winds, storm surges, heavy precipitation, and inland flooding (Pielke Jr et al., 2008; Mendelsohn et al., 2012; Klotzbach et al., 2018). It is, therefore, crucial to understand changes in intense TC characteristics (Elsner et al., 2008; Patricola & Wehner, 2018).”

References

Elsner, J., Kossin, J. & Jagger, T. The increasing intensity of the strongest tropical cyclones. *Nature* **455**, 92–95 (2008).

Klotzbach, P., Bowen, S., Pielke Jr. R. & Bell, M. Continental U.S. Hurricane Landfall Frequency and Associated Damage: Observations and Future Risks. *Bull. Am.*

Meteorological Soc. **99**, 1359–1376 (2018).

Mendelsohn, R., Emanuel, K., Chonabayashi, S. & Bakkensen, L. The impact of climate change on global tropical cyclone damage. *Nat. Clim. Change* **2**, 205–209 (2012).

Patricola, C. & Wehner, M. Anthropogenic influences on major tropical cyclone events. *Nature* **563**, 339–346 (2018).

Pielke Jr. R. et al. Normalized Hurricane Damage in the United States: 1900–2005. *Nat. Hazards Rev.* **9**, 29–42 (2008).

Line 51: By “other” do you mean less intense?

Response

Yes, it is less intense TCs. Then, following the referee’s comment, we have used ‘less intense TCs’ where appropriate throughout the revised manuscript.

Line 54: “TCs are more concentrated in summer” is vague. Are you referring to frequency here? And is this statement true across basins? The seasonal peak is often not during JJA (for the NH).

Response

Thank you for this comment. Indeed, TCs occur frequently during the middle-summer and early-autumn over most regions. Most of the intense TCs are concentrated in the autumn, later than the less intense TCs. The relevant statements have been modified accordingly.

Line 57: I don't think this is needed.

Response

This sentence has been removed accordingly.

Lines 61–65: A statement is needed at the start of this paragraph explaining that the paragraph discusses seasonality of different type of high-impact weather and the notion of compound risk.

Response

This is a valuable comment. We agree that an explanation of the seasonality of different types of high-impact weather events and the notion of compound risk at the start of this paragraph would greatly improve readability.

Following the referee's suggestion, we have modified this paragraph as follows, "In general, intense TCs occur more frequently in autumn than in summer because they require sufficient heat from the ocean to develop (Mei et al., 2015; Shan & Yu, 2020). The seasonal cycle of intense TCs lags behind that of other high-impact weather events (e.g., extreme rainfall events produced by the summer monsoon system), which often peak in summer and are largely determined by the seasonal cycle of atmospheric energy driven by solar radiation (Song et al., 2018, 2021). Recently, the compound hazards of intense TCs and other high-impact weather events have attracted increasing attention (Zscheischler et al., 2018; Matthews et al., 2019). The devastating impact of the compound hazards is well beyond any one of these events individually. They can induce substantial inland rain and flooding associated with multiple weather systems, causing large-scale failures of power and transportation systems, straining emergency responses, and depleting disaster preparation resources (Klotzbach et al., 2020; Feng et al., 2022; Patricola et al., 2022; Zhu et al., 2022). As the preferred time of occurrence of intense TCs and other high-impact weather events is usually off by one season, the likelihood of their simultaneous occurrence is generally assumed to be small; however, given the seasonal advance of intense TC occurrence, as shown in this study, its potential change should be considered."

In addition, we have taken the persistent rainfall event as an example of compound

hazards, which occurs during the overlap period between extreme rainfall produced by other weather systems (e.g., summer monsoon depressions) and that produced by intense TCs. The persistent rainfall event can lead to disproportionate impacts by inducing a superimposed elevation of water levels and a shortage of disaster preparation resources (Easterling et al. 2000; Chen & Zhai, 2013; Patricola et al., 2022). A detailed analysis of the persistent rainfall event has been added in the revised manuscript.

References

- Chen, Y. & Zhai, P. Persistent extreme precipitation events in China during 1951–2010. *Clim. Res.* **57**, 143–155 (2013).
- Easterling, D. et al. Climate extremes: observations, modeling, and impacts. *Science* **289**, 2068–2074 (2000).
- Feng, K. Ouyang, M. & Lin, N. Tropical cyclone-blackout-heatwave compound hazard resilience in a changing climate. *Nat. Commun.* **13**, 4421 (2022).
- Klotzbach, P. et al. Surface pressure a more skillful predictor of normalized hurricane damage than maximum sustained wind. *Bull. Am. Meteorological Soc.* **101**, E830–E846 (2020).
- Matthews, T., Wilby, R. & Murphy, C. An emerging tropical cyclone–deadly heat compound hazard. *Nat. Clim. Change* **9**, 602–606 (2019).
- Mei, W. et al. Northwestern Pacific typhoon intensity controlled by changes in ocean temperatures. *Sci. Adv.* **1**, e1500014 (2015).
- Patricola, C., Cassidy, D. & Klotzbach, P. Tropical oceanic influences on observed global tropical cyclone frequency. *Geophys. Res. Lett.* **49**, e2022GL099354 (2022).
- Shan, K. & Yu, X. Interdecadal variability of tropical cyclone genesis frequency in Western North Pacific and South Pacific Ocean basins. *Environ. Res. Lett.* **15**, 064030 (2020).
- Song, F., Leung, R., Lu, J. & Dong, L. Seasonally dependent responses of subtropical highs and tropical rainfall to anthropogenic warming. *Nat. Clim. Change* **8**, 787–792 (2018).

Song, F. et al. Emergence of seasonal delay of tropical rainfall during 1979–2019. *Nat. Clim. Change* **11**, 605–612 (2021).

Zhu, Y., Collins, J., Klotzbach, P. & Schreck III, C. Hurricane Ida (2021): rapid intensification followed by slow inland decay. *Bull. Am. Meteorological Soc.* **103**, E2354–E2369 (2022).

Zscheischler, J. et al. Future climate risk from compound events. *Nat. Clim. Change* **8**, 469–477 (2018).

Lines 86–87: It’s interesting that there is no change in the North Atlantic, only in the Gulf of Mexico. PI has increased in this basin (Klotzbach et al., 2022). The authors will need to discuss this further.

Response

We thank the referee for pointing this out. Indeed, there is a significant trend in the median value of the occurrence of intense TCs over the entire North Atlantic basin, with a rate of 0.8 days per decade (also shown in Extended Data Table 1).

To address the concerns of the referee, the climatological distribution of intense TCs and the inter-seasonal difference between the early and late seasons over the North Atlantic basin are given in Figure R6. It is found that intense TCs (as well as RI events) are mostly concentrated in the Gulf of Mexico and the western part of the North Atlantic basin (Supplementary Figure 1; also shown as Figure R6 here). In agreement with this, the inter-seasonal difference in intense TCs is much larger in the Gulf of Mexico and the western part of the North Atlantic basin. Furthermore, as shown in Figure R7 and Figure 5c by Klotzbach et al. (2022), an obvious increase in PI is observed in the Gulf of Mexico and the western part of the North Atlantic basin, while changes in PI in the eastern part are much weaker or even negative. These results indicate that the spatial pattern of seasonal changes in intense TCs over the North Atlantic basin is associated with the climatological location of intense TCs and the changes in thermodynamic factors.

To clarify the spatial pattern of inter-seasonal difference of intense TCs over the

North Atlantic basin, we have added a few sentences to the revised manuscript as follows: “Note that the significant earlier-shift trend of intense TC occurrence in the North Atlantic basin features large earlier shifts in the Gulf of Mexico and the western part of the North Atlantic basin and nearly no change in the eastern part, which is associated with the rarity of intense TC occurrence in the eastern part (Supplementary Figure 1) and changes in oceanic conditions (Klotzbach et al. 2022)”.

Figure R6. Maps of the occurrence frequency of intense TCs and the inter-seasonal difference between the early and late seasons. a, Distribution of the track density of intense TCs and the location of RI events over the North Atlantic basin during 1981–2017. **b**, Linear trends of the inter-seasonal difference in intense TC number between the early and late seasons. The black crosses indicate significance at the 95% confidence level. The annual number of intense TCs is calculated over $5^\circ \times 5^\circ$ boxes prior to computing trends.

Figure R7. Maps of 1990 – 2021 per-decade trend values (shaded) and annual trend p-values ≤ 0.10 (hatching) for annually averaged PI (Figure 5c by Klotzbach et al. (2022)).

References

Klotzbach, P. et al. Trends in global tropical cyclone activity: 1990–2021. *Geophys. Res. Lett.* **49**, e2021GL095774 (2022).

Lines 132–137: These results (Fig. 2b, e) are very interesting, but the authors should show the curves for both ADT-HURSAT and best-track data. The authors are correct that ADTHURSAT is more appropriate for trend analysis, but they should acknowledge that ADTHURSAT intensities may not be as accurate as the best-track data for the most intense TCs. In other words, ADT-HURSAT sacrifices intensity accuracy for homogenisation for trend analysis.

Response

Thank you for your insightful comments. Following your suggestion, we have repeated the analyses of seasonal changes in the occurrence of TC events with different intensities based on the suggested best-track data. In agreement with the ADT-HURSAT dataset, there is a significant earlier-shifting trend for intense TCs in both hemispheres, but not for less intense TC events (Extended Data Figure 1; Figure R8 here).

We believe that the inclusion of the best-track data will increase confidence that the results are not simply an artifact of the specific dataset utilized in the study. We have also added several sentences in the Methods to discuss the merits and caveats of the ADT-HURSAT and best-track datasets in trend detections.

Figure R8. Linear trends of the occurrence date of TC events with different intensities during the active season from 1981 to 2021 obtained from the best-track dataset in the (a) NH, and (b) SH. The shadings represent 95% confidence intervals in the linear regression analysis.

Lines 166–173: Bhatia et al. (2022), who show changes in global trends in RI, PI and TC environments based on the same dataset, is cited in the manuscript. This paper also discusses area-mean versus local TC environments, and some discussion of the two approaches would be a useful addition to the present manuscript. Reading through this submission, I think that some analysis of local TC environments (i.e., within a radius of TC centres) would be a worthwhile addition, if only in observations.

Response

This point is well-taken, and the revised version of the paper now explicitly includes results for the analysis of the favorable environmental conditions for TC RI process. We have calculated the environmental thresholds for RI events following the method by Bhatia et al. (2022). Consistent with Bhatia et al. (2022), we found that stronger PI, higher OHC, higher relative humidity, and weaker vertical wind shear are all beneficial for RI. The exact value differs somewhat from Bhatia et al. (2022), as we use 35kt as the threshold of RI definition rather than 30kt in Bhatia et al. (2022). This result supports that the high quartiles of PI, OHC and relative humidity and low quartiles of vertical wind shear could be applied to indicate the environmental conditions that lead to RI. In the main text, the 95th of PI, OHC, and relative humidity, and the 5th of vertical wind shear are used. We have also added a sensitivity analysis, in which the 90th of PI, OHC, and relative humidity, and the 10th of vertical wind shear are used. Consistently, an earlier shift of the seasonal cycle favorable oceanic conditions is obvious while no significant change in the atmospheric conditions. Our original conclusions still hold and these additional analyses have significantly improved the quality of the paper.

References

Bhatia, K. et al. A potential explanation for the global increase in tropical cyclone rapid intensification. *Nat. Commun.* **13**, 6626 (2022).

Line 203: Ext. Data Table 2 does not provide more information—see comment on tables below.

Response

We have added the information to Extended Data Table 2 accordingly.

Lines 205–208: Explain the large October decrease in panel e.

Response

Yes, we agree that the low vertical wind shear rate, which indicates favorable atmospheric conditions for RI events and is defined as the fractional proportion of the area with low percentiles of vertical wind shear relative to the tropical ocean area, exhibits a large decrease in October based on the ERA-5 reanalysis dataset (panel e in Extended Data Figure 6 in the first version of the manuscript; also shown as panel b in Extended Data Figure 5 in the revised manuscript and Figure R1 in the response letter).

However, the low vertical wind shear rate exhibits a much weaker and insignificant change in October when it is calculated based on the JRA-55 (Extended Data Figure 6 in the manuscript; also shown as Figure R2 in the response letter) and MERRA-2 reanalysis datasets (Extended Data Figure 7 in the manuscript; also shown as Figure R3 in the response letter). Hence, the inconsistent variations in vertical wind shear between different reanalysis datasets are largely due to the uncertainty of the reanalysis datasets.

We appreciate the referee for pointing out the importance of vertical wind shear in moderating TC seasonal activity. However, in the absence of consistency in the seasonal changes of vertical wind shear among these three atmospheric reanalysis datasets to date, we leave it as a future study.

Line 262. “will be further amplified under global warming”—this is likely true, but the authors have not shown, for example, that in control simulations without anthropogenic factors, seasonality shifts do not occur. Analysis of control simulations would be a necessary basis on which to make attribution-type statements.

Response

Thank you for your insightful comments. We have taken your suggestions and in tandem with the suggestions of Referee #1, and we have made great efforts to conduct additional analyses to better support the detection and attribution conclusions. Firstly, the seasonal changes of the high percentiles of PI and OHC have been estimated based on the CESM2 large ensemble, by which internal variability could be largely suppressed. In Figure R9, the high percentiles of PI and OHC show significant trends towards earlier onset, consistent with these based on the multi-model mean of the CMIP6 historical simulations. To make a stronger attribution statement, we have repeated the analysis above based on the DAMIP experiments under the greenhouse gas (GHG) forcing only, natural (NAT) forcing only, and anthropogenic aerosol (AER) forcing only, respectively. Results suggest that the earlier onset of high percentiles of PI and OHC is primarily driven by GHG forcing, while the contributions of the NAT and AER forcings are negligible or even negative (Figure R10). The results, method, and figures are updated. Please see the abstract for a summary of our detection and attribution statements, “Using simulations from multiple global climate models, large ensemble, and individual forcing experiments, the earlier onset of favorable oceanic conditions is detectable and is found to be primarily driven by greenhouse gas forcing.”

Overall, the inclusion of the additional analyses of the CESM2 large ensemble simulations and DAMIP experiments has strengthened the relationships between the seasonal advance of intense TCs and anthropogenic warming.

Figure R9. Changes in the seasonal cycle of the oceanic conditions obtained from the CMIP6 historical simulations and the CESM2 large ensemble. For (up) the multi-model mean of the CMIP6 historical simulations and (bottom) the CESM2 large ensemble. **a, b,** The CMIP6 multi-model mean of the climatological value of PI (solid line) and the linear trend of high PI rate (bar; the fractional proportion of area with the PI value above its 95th percentile relative to the tropical ocean area) in each month during the active season from 1981 to 2017 in the **(a)** NH, and **(b)** SH. The error bars indicate the standard deviation. **c, d,** The CMIP6 multi-model mean of the climatological value of OHC (solid line) and the linear trend of high OHC rate (bar; the fractional proportion of area with the OHC value above its 95th percentile relative to the tropical ocean area) in each month during the active season from 1981 to 2017 in the **(c)** NH, and **(d)** SH. **e–h,** Same as in **a–d,** respectively, but obtained from the CESM2 large ensemble.

Figure R10. Changes in the seasonal cycle of oceanic conditions obtained from the DAMIP experiments under the greenhouse gas (GHG) forcing only, natural (NAT) forcing only, and anthropogenic aerosol (AER) forcing only. For (up) the multi-model mean of experiments forced by the GHG, (middle) experiments forced by the NAT, and (bottom) experiments forced by the AER. **a, b**, The climatological value of PI (solid line) and the linear trend of high PI rate (bar) in each month obtained from the GHG experiments during the active season from 1981 to 2017 in the (a) NH, and (b) SH. The error bars indicate the standard deviation. **c, d**, The climatological value of OHC (solid line) and the linear trend of high OHC rate (bar) in each month obtained from the GHG experiments during the active season from 1981 to 2017 in the (c) NH, and (d) SH. **e–h**, Same as in **a–d**, respectively, but obtained from the NAT experiments. **i–l**, Same as in **a–d**, respectively, but obtained from the AER experiments.

Line 341: Change to Ext. Data Table 2.

Response

We have changed it to Extended Data Table 2 accordingly.

Comments on figures and tables

Fig 1c. The cross symbols are quite coarse. Please replace with finer-scale hatching to show exactly where is significant more clearly.

Response

Thank you for the suggestion. We originally used the grid with a $10^\circ \times 10^\circ$ resolution, and we have changed to a finer grid with a $5^\circ \times 5^\circ$ resolution per your suggestion (Figure R11). This change has resulted in slight differences in trends and significance, but there are no major differences in the key results. The figure and caption are updated.

Figure R11. Linear trends of the inter-seasonal difference in intense TC number between the early and late seasons. The early season is defined as June–August in the NH and December–February in the SH. The late season is defined as September–November in the NH and March–April in the SH. The black crosses indicate significance at the 95% confidence level. The annual number of intense TCs is calculated over $5^\circ \times 5^\circ$ boxes prior to computing trends.

Fig. 2. Show the trend uncertainty in panels a, c, d, f.

Response

Thank you for the suggestion. We have added shadings to show the uncertainty of the trends in Fig. 2 a, c, d, f.

Ext. Data Table 2. This does not offer much information. Please add at least these details about each model: ocean and atmosphere resolution, ocean model, mean TC activity, references.

Response

We have included the information per your suggestion.

Once again, thank you for your comments and suggestions, and for your contributions to improving our paper.

Reviewer Reports on the First Revision:

Referees' comments:

Referee #1 (Remarks to the Author):

Review of Seasonal Advance of Intense Tropical Cyclones in A Warming Climate by Shan et al.

In my previous review of this article, my main concern was that some of the conclusions seemed a bit overstated and some methods needed to be clarified. In particular, the biggest issues were that there needed to be

1. A clearer comparison to a closely related prior study (Truchelut et al 2022).
2. A more formal attribution of the changes in oceanic condition to external forcing.
3. A softening of the conclusion regarding compound events.
4. An addition of more modern reanalysis than NCEP/NCAR.

The authors have performed significant revisions to address these and other concerns, including replacing the NCEP/NCAR reanalysis with JRA-55 and MERRA-2 and estimating the seasonal changes of favorable oceanic conditions with the CESM2 large ensemble and single forcing experiments. These changes strengthen the study and make it more suitable for publication. In general, my concerns have been satisfied, though I have a few remaining minor comments. The most substantial is #4; I don't know why the authors haven't also calculated PI with the 3 atmospheric reanalyses.

1. Line 395: It is still not clear how the favorable conditions for RI are quantified. The sentence reads "The favorable conditions are quantified by the rate of the area with the high PI...", but what is meant by "the rate of the area"? Do you mean that "The trends in the favorable conditions are quantified by the rate of change in the area that has high PI..."? That is...if the area with high PI etc... increases, that means that the favorability for RI has increased?

2. Line 392: this is an incomplete sentence.

3. Lines 191, 369-371, 380-382: You state that you calculate the oceanic factors, including PI, using the ECMWF Ocean Reanalysis database and the GODAS. But the PI calculation requires profiles of atmospheric temperature and specific humidity in addition to sea surface temperature...where do the atmospheric variables for the PI calculation come from?

4. Lines 380-382, 383-386: In addition to calculation the atmosphere factors (relative humidity and vertical wind shear) in the three different reanalyses, you should also calculate PI with ERA-5, JRA-55, and MERRA-2.

5. Line 206, Figure 3 caption: Which dataset is presented as "observed results (OBS)"?

6. Figure 3: What are the units here? They are stated in the label as "trend of high PI rate (decade-1)" but can you be more specific about what that means? Is it the change in the fractional area covered by high PI per decade?

7. Line 197: I suggest revising this to read "While atmospheric factors (i.e., relative humidity and vertical wind shear) were previously documented to also impact RI events and are widely applied as forecasting variables..."

8. Line 192: I think this should say "...by the area with the oceanic factor above its 95th percentile" rather than up to its 95th percentile.

9. Line 45, 150-152: The statement that the Truchelut et al 2022 study found no significant seasonal

change in total Atlantic TC activity is misleading because Truchelut et al 2022 DID find a significant trend towards earlier onset of the North Atlantic TC season. To mean, that seems like a seasonal change in Atlantic TC activity! I think you just need to be more explicit about what you referring to. They found that there was no trend in the date that Atlantic ACE reached certain thresholds beyond the initial percentiles.

10. Figure 2: What are the units of panel b and e? It is sated as “trend of date (decade-1)”; is it meant to be a trend with units of days per decade? Please specify in the plot and caption.

11. I’m a little confused about the shading and significance in Figure 2. The open/filled circles indicate the 95% significance based on a non-parametric test, but then you also include the 95% confidence intervals in shading. In panel 2, for the intensity around 110 kt, the dot indicates this as significance, but the shading goes far above a 0 trend. I know they are two different statistical calculations, but it doesn’t make sense for them to be so inconsistent.

12. Line 88: I suggest saying “...annual number in each month from 1981-2017” to make clear you are summing over all months.

13. Figure 1: What are the units of the trend in panels a and b? Is it number of storms per decade? Please specify in the plot and caption.

14. Line 181, 190, 394-397: It make sense that these particular environmental factors are relevant for RI, but it still isn’t clear to me why specifically it is the fractional area of the basin that has these factors above certain threshold that matters. Wouldn’t the local conditions experienced by TCs be more relevant for the likelihood that they undergo RI than the fraction of the overall basin that has a favorable environment? I know the authors have refined their analysis of how they calculate the thresholds by following the method of Bhatia et al (2022), but they should also at least mention the possibility that the local environment is what instead should be considered. To be clear, I’m not disputing that high percentiles of these environmental factors are beneficial for RI. I’m just asking for clarification about why the authors use the fractional area coverage as the metric of favorability.

Referee #2 (Remarks to the Author):

The authors have thoroughly addressed the comments I made reviewing the first submission. As a result, I now recommend publication, but, to reiterate a comment from my previous review, the work is perhaps better suited to Nature Climate Change, although not unsuited to Nature.

Author Rebuttals to First Revision:

Author Response to Referee#1 of “Seasonal Advance of Intense Tropical Cyclones in A Warming Climate”

General Comments

In my previous review of this article, my main concern was that some of the conclusions seemed a bit overstated and some methods needed to be clarified. In particular, the biggest issues were that there needed to be

1. A clearer comparison to a closely related prior study (Truchelut et al 2022).
2. A more formal attribution of the changes in oceanic condition to external forcing.
3. A softening of the conclusion regarding compound events.
4. An addition of more modern reanalysis than NCEP/NCAR.

The authors have performed significant revisions to address these and other concerns, including replacing the NCEP/NCAR reanalysis with JRA-55 and MERRA-2 and estimating the seasonal changes of favorable oceanic conditions with the CESM2 large ensemble and single forcing experiments. These changes strengthen the study and make it more suitable for publication. In general, my concerns have been satisfied, though I have a few remaining minor comments. The most substantial is #4; I don't know why the authors haven't also calculated PI with the 3 atmospheric reanalyses.

Response

Thanks for the encouragement. We appreciate the insight and thoughtfulness of the referee's comments and suggestions and will attempt to address each of them fully here. Indeed, only the ERA-5 dataset was used to calculate potential intensity (PI) in the previous manuscript. Following your suggestion, we have added the PI results based on the JRA-55 and MERRA-2 datasets in the revised manuscript. The earlier onset of the fractional area covered by high PI also occurs in the JRA-55 and MERRA-2 datasets, in agreement with that based on the ERA-5 dataset (Figure R1).

Figure R1. Changes in the fractional area covered by high PI based on different reanalyses. For (left) with the oceanic variables from the ECMWF and (right) GODAS datasets. For (up) with the atmospheric variables from the ERA-5, (middle) JRA-55, and (bottom) MERRA-2 datasets. **a, b**, The climatological value of PI (solid line) and the linear trend of the fractional area covered by high PI during the active season during 1981–2017 in the (a) NH, and (b) SH based on the oceanic variables from the ECMWF dataset and the atmospheric variables from the ERA-5 dataset. The error bars indicate the standard deviation. **c-d**, Same as in **a-b**, respectively, but based on the oceanic variables from the GODAS dataset. **e-h**, Same as in **a-d**, respectively, but based on the atmospheric variables from the JRA-55 dataset. **i-l**, Same as in **a-d**, respectively, but based on the atmospheric variables from the MERRA-2 dataset.

Minor Comments

1. Line 395: It is still not clear how the favorable conditions for RI are quantified. The sentence reads “The favorable conditions are quantified by the rate of the area with the high PI...”, but what is meant by “the rate of the area”? Do you mean that “The trends in the favorable conditions are quantified by the rate of change in the area that has high PI...”? That is...if the area with high PI etc... increases, that means that the favorability for RI has increased?

Response

Thank you for catching this. We have taken your suggestions (Comment 6) and rewrite the relevant statements as follows: “Here, the favorable conditions are quantified by the fractional area covered by high PI, high OHC, high RH, and low VWS relative to the overall tropical ocean area (5° N-30° N, 5°S-30° S)”.

2. Line 392: this is an incomplete sentence.

Response

Thanks! We have revised this sentence and checked the manuscript to ensure that it is typo free.

3. Lines 191, 369-371, 380-382: You state that you calculate the oceanic factors, including PI, using the ECMWF Ocean Reanalysis database and the GODAS. But the PI calculation requires profiles of atmospheric temperature and specific humidity in addition to sea surface temperature...where do the atmospheric variables for the PI calculation come from?

Response

Thanks for raising this issue. The ERA-5 dataset was used for PI calculation in the previous version of the manuscript. The relative statements have been revised to clarify

this. In the new version of the manuscript, following your suggestion, two oceanic reanalysis datasets and three atmospheric reanalysis datasets have been used for PI calculation.

4. Lines 380-382, 383-386: In addition to calculation the atmosphere factors (relative humidity and vertical wind shear) in the three different reanalyses, you should also calculate PI with ERA-5, JRA-55, and MERRA-2.

Response

Thanks for pointing this out. Following your suggestion, we have estimated the seasonal changes of the fractional area covered by high PI based on the ERA-5, JRA-55, and MERRA-2 (Figure R1). Consistent with the results based on the ERA-5 reanalysis dataset, the fractional area covered by high PI shows significant trends towards earlier onset based on the JRA-55 and MERRA-2 reanalysis datasets. These results are reflected in Extended Data Figure 5 in the revised manuscript. Thank you for this useful suggestion, as the inclusion of the JRA-55 and MERRA-2 datasets has increased the confidence in our PI analysis.

5. Line 206, Figure 3 caption: Which dataset is presented as “observed results (OBS)”?

Response

The PI calculation in Figure 3 is based on the ECMWF and ERA5 datasets and the OHC calculation is based on the ECMWF dataset. The caption of Figure 3 has been revised to clarify this.

6. Figure 3: What are the units here? They are stated in the label as “trend of high PI rate (decade-1)” but can you be more specific about what that means? Is it the change in the fractional area covered by high PI per decade?

Response

Thanks! Yes, it is the change in the fractional area covered by high PI per decade. Following your suggestion, this unit has been used where appropriate throughout the revised manuscript.

7. Line 197: I suggest revising this to read “While atmospheric factors (i.e., relative humidity and vertical wind shear) were previously documented to also impact RI events and are widely applied as forecasting variables...”

Response

Thank you for the suggestion. Revision has been made accordingly.

8. Line 192: I think this should say “...by the area with the oceanic factor above its 95th percentile” rather than up to its 95th percentile.

Response

Thanks! Revision has been made accordingly.

9. Line 45, 150-152: The statement that the Truchelut et al 2022 study found no significant seasonal change in total Atlantic TC activity is misleading because Truchelut et al 2022 DID find a significant trend towards earlier onset of the North Atlantic TC season. To mean, that seems like a seasonal change in Atlantic TC activity! I think you just need to be more explicit about what you referring to. They found that there was no trend in the date that Atlantic ACE reached certain thresholds beyond the initial percentiles.

Response

Thanks for this suggestion. The relevant statements have been revised accordingly.

10. Figure 2: What are the units of panel b and e? It is sated as “trend of date (decade-1)””; is it meant to be a trend with units of days per decade? Please specify in the plot and caption.

Response

Yes, it is the trend with units of days per decade. Following your suggestion, we have used this unit where appropriate throughout the revised manuscript.

11. I’m a little confused about the shading and significance in Figure 2. The open/filled circles indicate the 95% significance based on a non-parametric test, but then you also include the 95% confidence intervals in shading. In panel 2, for the intensity around 110 kt, the dot indicates this as significance, but the shading goes far above a 0 trend. I know they are two different statistical calculations, but it doesn’t make sense for them to be so inconsistent.

Response

Thank you for pointing this out. We have carefully checked the results to ensure they are correct. In panel 2 of Figure 2, for TCs with intensity around 110 kt, the earlier-shifting trend in the NH obtained from the ADT-HURSAT dataset is statistically

significant based on the two different statistical calculations, while the 95% confidence intervals for the SH are very large and go above a 0 trend. However, when we consider the TCs with intensity around 110 kt obtained from the best-track dataset as shown in Extended Data Figure 2, the earlier-shifting trends in the NH and SH obtained are statistically significant based on different statistical tests. Additionally, the earlier-shifting trends of TCs with intensity around 120 kt obtained from the two datasets are statistically significant in both the NH and the SH based on the different statistical calculations (Figure 2b, e; Extended Data Figure 2). These results suggest that the seasonal advance is more evident for more intense TCs as we emphasized here. Additional explanations have been included in the revised manuscript.

12. Line 88: I suggest saying “...annual number in each month from 1981-2017” to make clear you are summing over all months.

Response

Thanks! Revision has been made accordingly.

13. Figure 1: What are the units of the trend in panels a and b? Is it number of storms per decade? Please specify in the plot and caption.

Response

Thank you for the suggestion. Yes, it is the number of intense TCs per decade. We have revised the figures and captions where appropriate throughout the revised manuscript.

14. Line 181, 190, 394-397: It make sense that these particular environmental factors are relevant for RI, but it still isn't clear to me why specifically it is the fractional area of the basin that has these factors above certain threshold that matters. Wouldn't the local conditions experienced by TCs be more relevant for the likelihood that they undergo RI than the fraction of the overall basin that has a favorable environment? I know the authors have refined their analysis of how they calculate the thresholds by following the method of Bhatia et al (2022), but they should also at least mention the possibility that the local environment is what instead should be considered. To be clear, I'm not disputing that high percentiles of these environmental factors are beneficial for RI. I'm just asking for clarification about why the authors use the fractional area coverage as the metric of favorability.

Response

Thanks for bringing up this excellent point. We totally agree that the local conditions experienced by TCs should be more relevant for the likelihood that they undergo RI. However, it cannot be obtained with good accuracy in the low-resolution climate model simulations in which the TC/RI process cannot be represented well (e.g., historical and single forcing experiments). Instead, the fractional area coverage by high percentile PI and OHC, also corresponding well to RI events, is simple and can be directly calculated from the simulations. Hence, the fractional area coverage is used as the metric of favorability here. We thank you for pointing out the importance of the local environment in indicating the likelihood that they undergo RI. We have also pointed this out in the revised manuscript.

Reviewer Reports on the Second Revision:

Referees' comments:

Referee #1 (Remarks to the Author):

The authors have addressed all my concerns and I now recommend publication.